# Delving into Out-of-Distribution Detection with Vision-Language Representations

**Yifei Ming**[1]    **Ziyang Cai**[1]    **Jiuxiang Gu**[2]    **Yiyou Sun**[1]    **Wei Li**[3]    **Yixuan Li**[1]
[1]Department of Computer Sciences, University of Wisconsin-Madison
[2]Adobe    [3]Google Research
{alvinming,ziyangc,sunyiyou,sharonli}@cs.wisc.edu
jigu@adobe.com    mweili@google.com

## Abstract

Recognizing out-of-distribution (OOD) samples is critical for machine learning systems deployed in the open world. The vast majority of OOD detection methods are driven by a single modality (*e.g.*, either vision or language), leaving the rich information in multi-modal representations untapped. Inspired by the recent success of vision-language pre-training, this paper enriches the landscape of OOD detection from a single-modal to a multi-modal regime. Particularly, we propose Maximum Concept Matching (MCM), a simple yet effective zero-shot OOD detection method based on aligning visual features with textual concepts. We contribute in-depth analysis and theoretical insights to understand the effectiveness of MCM. Extensive experiments demonstrate that MCM achieves superior performance on a wide variety of real-world tasks. MCM with vision-language features outperforms a common baseline with pure visual features on a hard OOD task with semantically similar classes by 13.1% (AUROC). Code is available at https://github.com/deeplearning-wisc/MCM.

## 1 Introduction

Out-of-distribution (OOD) detection is critical for deploying machine learning models in the wild, where samples from novel classes can naturally emerge and should be flagged for caution. Despite increasing attention, the vast majority of OOD detection methods are driven by single-modal learning [26, 29, 34, 68, 89, 93, 95, 98]. For example, labels are typically encoded as one-hot vectors in image classification, leaving the semantic information encapsulated in texts largely unexploited. OOD detection relying on pure visual information can inherit the limitations, *e.g.*, when an OOD input may be visually similar to in-distribution (ID) data yet semantically different from any ID class.

In this paper, we delve into a new landscape for OOD detection, departing from the classic single-modal toward a *multi-modal* regime. While the motivation is appealing, a core challenge remains: *how to effectively utilize joint vision-language features for OOD detection?* In the visual domain, existing methods typically require good feature representations [66, 72], and a distance metric under which OOD data points are relatively far away from the in-distribution (ID) data [42, 71]. These approaches, however, do not directly translate into the multi-modal regime. On the representation learning side, recent vision-language pre-training schemes such as CLIP [59] and ALIGN [33] have emerged as promising alternatives for visual representation learning. The main idea is to align an image with its corresponding textual description in the feature space. While the resulting representations are powerful, OOD detection based on such aligned multi-modal features is still in its infancy.

We bridge the gap by exploring a distance-based OOD detection approach, leveraging the joint vision-language representations. Our method capitalizes on the compatibility between visual features and textual features. By defining the textual features as the "*concept prototypes*" for each ID class,

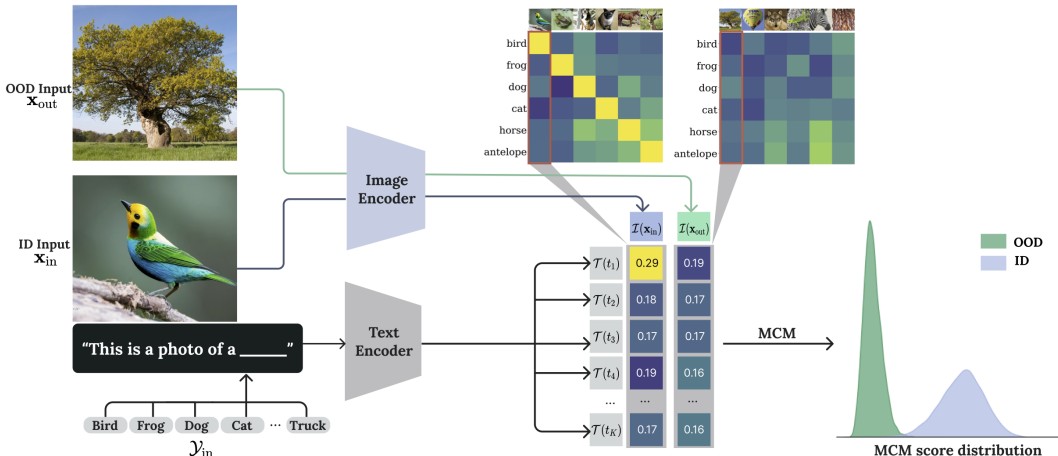

Figure 1: Overview of the proposed zero-shot OOD detection framework. The ID classification task is defined by a set of class labels $\mathcal{Y}_{in}$. The goal of OOD detection is to detect samples that do not belong to $\mathcal{Y}_{in}$. We view the textual embeddings of ID classes (wrapped by text templates) as concept prototypes. The OOD uncertainty of an input image can be characterized by the distance from visual features to the closest ID prototype. By properly scaling the distance, the MCM score achieves strong ID-OOD separability. See Section 3 for details.

we characterize OOD uncertainty by the distance from the visual feature to the closest ID prototype. That is, images closer to one of the textual embeddings of ID classes are more likely to be ID and vice versa. By a proper scaling of the distance, our proposed Maximum Concept Matching (**MCM**) score achieves strong ID-OOD separability (see Figure 1). MCM stands in contrast with the previous distance-based approaches, such as Mahalanobis [42], which defines class prototypes based on pure visual embeddings. Indeed, we show later in Section 5 that MCM (with multi-modal vision-language features) is far more competitive than Mahalanobis (with single-modal visual features). Moreover, while prior works of CLIP-based OOD detection [16, 19] rely on a set of candidate OOD labels, MCM is OOD-agnostic and alleviates the need for any prior information about test inputs.

Our work also advances the field by showcasing the promise of zero-shot OOD detection, which offers strong performance and generality without training on the ID samples. In particular, classic OOD detection methods often require training from scratch [9, 27] or fine-tuning [19, 32] on a given ID dataset. In this setting, a classifier and its companion OOD detector are good at only one task. Every new task (ID dataset) requires additional training and brings additional computation and storage costs. In contrast, we show for the first time that: **(1)** MCM achieves superior performance across a wide variety of real-world tasks—with just *one single pre-trained model*. This is encouraging given that there is no training or any OOD information involved. **(2)** On the challenging ImageNet-1k benchmark, MCM's zero-shot OOD detection performance favorably matches and even outperforms strong task-specific baselines fine-tuned on BiT [32] and ViT models [19]. **(3)** MCM remains robust against hard OOD inputs, including both semantically hard OODs [85] and spurious OODs [50].

We summarize our main contributions as follows:

1. We propose MCM, a simple yet effective OOD detection method based on aligned vision-language features. MCM offers several compelling advantages over other OOD detection methods: generalizable (one model supports many tasks), OOD-agnostic (no information required from OOD data), training-free (no downstream fine-tuning required), and scalable to large real-world tasks.

2. We conduct extensive experiments and show that MCM achieves superior performance on a wide range of real-world tasks. On ImageNet-1k, MCM achieves an average AUROC of 91.49%, outperforming methods that require training. Moreover, we show that MCM remains competitive under challenging hard OOD evaluation tasks.

3. We provide in-depth empirical and theoretical analysis, providing insights to understand the effectiveness of MCM. We hope that this work will serve as a springboard for future works on OOD detection with multi-modal features.

## 2 Preliminaries

**Contrastive vision-language pre-training.** Compared to visual representation learning models such as ViT [13], vision-language representation learning demonstrates superior performance on image classification tasks. For instance, CLIP [59] adopts a self-supervised contrastive objective (*i.e.*, InfoNCE loss [75]) to align an image with its corresponding textual description in the feature space. Specifically, CLIP adopts a simple dual-stream architecture with one text encoder $\mathcal{T} : t \to \mathbb{R}^d$ (*e.g.*, Transformer [77]) and one image encoder $\mathcal{I} : \mathbf{x} \to \mathbb{R}^d$ (*e.g.*, ViT [13]). After pre-training on a dataset of 400 million text-image pairs, the joint vision-language embeddings of CLIP well associate objects in different modalities. Despite the promise, existing CLIP-like models perform zero-shot classification in a *closed-world* setting. That is, it will match an input into a fixed set of categories, even if it is irrelevant (*e.g.*, a tree being predicted as a bird in Figure 1). This motivates our work to leverage the multi-modal representation for OOD detection, which is largely unexplored.

**Zero-shot OOD detection.** Given a pre-trained model, a classification task of interest is defined by a set of class labels/names $\mathcal{Y}_{\text{in}}$, which we refer to as the known (ID) classes. Here ID classes are defined *w.r.t.* the classification task of interest, instead of the classes used in pre-training. Accordingly, OOD is defined *w.r.t.* the ID classes, not the data distribution during pre-training. The goal of OOD detection is to (1) detect samples that do not belong to any of the known classes; (2) otherwise, assign test samples to one of the known classes. Therefore, the OOD detector can be viewed as a "safeguard" for the classification model. Formally, we denote the OOD detector as a binary function: $G(\mathbf{x}; \mathcal{Y}_{\text{in}}, \mathcal{T}, \mathcal{I}) : \mathcal{X} \to \{\text{in}, \text{out}\}$, where $\mathbf{x} \in \mathcal{X}$ denotes a test image. Our method is based on only the names of the given classes in $\mathcal{Y}_{\text{in}}$, and a pre-trained model. Different from standard supervised learning, there is no training on the ID samples involved, hence zero-shot.

## 3 OOD Detection via Concept Matching

We illustrate our approach in Figure 1, which derives the OOD detector $G(\cdot)$ based on *concept matching*. For a given task with label set $\mathcal{Y}_{\text{in}} = \{y_1, y_2, ..., y_K\}$, we can construct a collection of concept vectors $\mathcal{T}(t_i), i \in \{1, 2, ..., K\}$, where $t_i$ is the text prompt "`this is a photo of a` $\langle y_i \rangle$" for a label $y_i$. The concept vectors are represented by the embeddings of the text prompts.

For any test input image $\mathbf{x}'$, we can calculate the label-wise matching score based on the cosine similarity between the image feature $\mathcal{I}(\mathbf{x}')$ and the concept vector $\mathcal{T}(t_i)$: $s_i(\mathbf{x}') = \frac{\mathcal{I}(\mathbf{x}') \cdot \mathcal{T}(t_i)}{\|\mathcal{I}(\mathbf{x}')\| \cdot \|\mathcal{T}(t_i)\|}$. Formally, we define the maximum concept matching (**MCM**) score as:

$$S_{\text{MCM}}(\mathbf{x}'; \mathcal{Y}_{\text{in}}, \mathcal{T}, \mathcal{I}) = \max_i \frac{e^{s_i(\mathbf{x}')/\tau}}{\sum_{j=1}^{K} e^{s_j(\mathbf{x}')/\tau}}, \tag{1}$$

where $\tau$ is the temperature. For ID data, it will be matched to one of the concept vectors (textual prototypes) with a high score; and vice versa. Formally, our OOD detection function can be formulated as:

$$G(\mathbf{x}'; \mathcal{Y}_{\text{in}}, \mathcal{T}, \mathcal{I}) = \begin{cases} 1 & S_{\text{MCM}}(\mathbf{x}'; \mathcal{Y}_{\text{in}}, \mathcal{T}, \mathcal{I}) \geq \lambda \\ 0 & S_{\text{MCM}}(\mathbf{x}'; \mathcal{Y}_{\text{in}}, \mathcal{T}, \mathcal{I}) < \lambda \end{cases},$$

where by convention 1 represents the positive class (ID) and 0 indicates OOD. $\lambda$ is chosen so that a high fraction of ID data (*e.g.*, 95%) is above the threshold. For samples that are classified as ID, one can obtain the class prediction based on the closest concept: $\hat{y} = \arg\max_{i \in [K]} s_i$.

**Remark**: (1) Our work differs from (and is complementary to) CLIP by focusing on OOD detection rather than (closed-world) zero-shot classification. We show new theoretical insights that softmax scaling plays a unique role in zero-shot OOD detection—improving the separability between ID and OOD data. This role has not been studied rigorously for zero-shot OOD detection. Readers familiar with CLIP may notice that MCM can be used for zero-shot classification in the closed world. This also makes MCM practically convenient for dual goals: detect OOD samples and assign ID data to one of the known classes. (2) Our method in principle is not limited to CLIP; it can be generally applicable for contrastive vision-language pre-training models that promote multi-modal feature alignment.

**New insights on softmax scaling for zero-shot OOD detection.** We provide theoretical justifications that softmax scaling improves the separability between ID and OOD data for CLIP-based OOD detection, which is *contrary* to models trained with cross-entropy (CE) loss. In particular, CLIP-like models are trained with a multi-modal contrastive loss, which maximizes the cosine similarity between an image and its textual description in the feature space. The resulting cosine similarity scores display strong *uniformity*[1] across labels, as evidenced in Figure 2 (right). Compared to OOD inputs, the gap between the maximum cosine similarity and the average is larger for ID inputs. However, the gap can be small when the number of ID classes increases where ID samples occur with lower highest cosine similarity. As a result, the highest cosine similarity for ID samples and OOD samples can be highly close (*c.f.* Figure 2 (left)).

Motivated by these observations, MCM employs softmax as a post hoc mechanism to **magnify** the difference. This is *fundamentally different from the softmax score derived from a model trained with cross-entropy loss*, which inherently maximizes the posterior $p(y|\mathbf{x})$ for the ground-truth label, and minimizes the probability for other labels. Unlike CLIP-like models, logit scores displaying uniformity would be heavily penalized by the CE loss. As a result, the logit score corresponding to the ground-truth label can already be significantly higher than other labels. Applying softmax on the logit scores can exacerbate overconfident predictions, and reduce the separability between ID and OOD data [46]. Indeed, for a

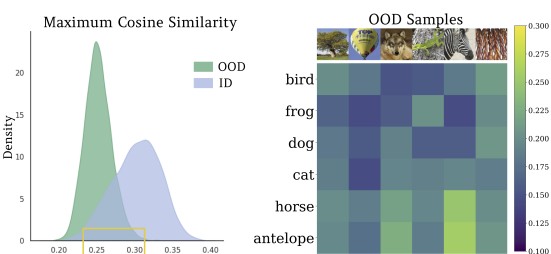

Figure 2: Left: Maximum cosine similarity for ID and OOD inputs. There exists overlapping regions (shown in yellow); Right: Cosine similarities between OOD inputs and ID concept vectors. For OOD inputs, the cosine similarities display uniformity.

model trained with cross-entropy loss, a logit-based score such as Energy [48] is shown to be much more effective than the softmax score.

Interestingly, for CLIP-like models, the trend is the opposite—applying softmax helps sharpen the uniform-like inner product scores, and increases the separability between ID and OOD data. To help readers better understand the insights, we first formalize our observations that OOD inputs trigger *similar cosine similarities* across ID concepts (Figure 2, right) as the following assumption:

**Assumption 3.1.** Let $z := \mathbb{1}\{y \in \mathcal{Y}_{\text{in}}\}$. $Q_{\mathbf{x}}$ denotes the out-of-distribution $\mathbb{P}_{\mathbf{x}|z=0}$ (marginal distribution of $\mathbf{x}$ conditioned on $z = 0$). Assume $\exists \delta > 0$ such that

$$Q_{\mathbf{x}}\left(\frac{1}{K-1}\sum_{i \neq \hat{y}}[s_{\hat{y}_2}(\mathbf{x}) - s_i(\mathbf{x})] < \delta\right) = 1,$$

where $\hat{y} := \text{argmax}_{i \in [K]} s_i(\mathbf{x})$ and $\hat{y}_2 := \text{argmax}_{i \neq \hat{y}, i \in [K]} s_i(\mathbf{x})$ denote the indices of the largest and second largest cosine similarities for an OOD input $\mathbf{x}$.

Now we provide formal guarantees that using softmax can provably reduce the false positive rate (FPR) compared to that without softmax.

**Theorem 3.1.** Given a task with ID label set $\mathcal{Y}_{\text{in}} = \{y_1, y_2, ..., y_K\}$ and a pre-trained CLIP-like model $(\mathcal{T}, \mathcal{I})$. If $Q_{\mathbf{x}}$ satisfies Assumption 3.1, then there exists a constant $T = \frac{\lambda(K-1)(\lambda^{\text{wo}}+\delta-s_{\hat{y}_2})}{K\lambda-1}$ such that for any temperature $\tau > T$, we have

$$\text{FPR}(\tau, \lambda) \leq \text{FPR}^{\text{wo}}(\lambda^{\text{wo}}),$$

where $\text{FPR}(\tau, \lambda)$ is the false positive rate based on softmax scaling *with* temperature $\tau$ and detection threshold $\lambda$; $\text{FPR}^{\text{wo}}(\lambda^{\text{wo}})$ is the false positive rate *without* softmax scaling based on threshold $\lambda^{\text{wo}}$.

---

[1]This can be explained both theoretically [84] and empirically [81]. It has been shown that self-supervised contrastive learning with a smaller temperature (*e.g.*, initialized as 0.07 for CLIP) promotes uniform distribution for $L_2$-normalized features. Moreover, as CLIP features lie on a high-dimensional space (512 for CLIP-B/16 and 768 for CLIP-L/14), uniformly distributed points in a high-dimensional sphere tend to be equidistant to each other [79]. Therefore, for OOD inputs, we observe approximately uniform cosine similarity with concept vectors.

This suggests that applying softmax scaling with a moderate temperature results in superior OOD detection performance compared to that without softmax scaling. The proof is in Appendix A. Later in Section 5, we empirically verify on a real-world ImageNet dataset that our bound can indeed be satisfied in CLIP where the thresholds are chosen at 95% true positive rate.

**What MCM offers:** Beyond theoretical insights, we would like to highlight several compelling advantages of our zero-shot OOD detection approach, owing to the strong pre-trained CLIP model:

- **Generalizable to many tasks**: Traditional OOD detection methods are based on a task-specific model. As a result, the OOD detector is not suitable for a realistic online scenario where the task changes from one to another. In contrast, we will show in Section 4 that MCM can perform a wide variety of OOD detection tasks, with just one single model. For a new task, only the names of the task's visual concepts $\mathcal{Y}_{in}$ are required.

- **OOD-agnostic**: Our method does not rely on any OOD information, and thus suits many real-world scenarios where one cannot anticipate what the unknowns would be ahead of time. This also mitigates the shortcoming of a recent approach [19], which assumes that a set of unseen labels are given as some weak information about OOD data.

- **Training-free**: MCM enables OOD detection in a zero-shot fashion. This stands in contrast to the vast majority of OOD detection literature, which often requires training from scratch or fine-tuning to achieve competitive performance.

- **Scalable**: The contrastive vision-language pre-training paradigm makes MCM scalable to a large number of class labels and realistic high-resolution images.

We now proceed to the experimental results, demonstrating these advantages on real-world tasks.

## 4 A Comprehensive Analysis of MCM

### 4.1 Datasets and Implementation Details

**Datasets.** Most previous works on OOD detection only focus on small-scale datasets with blurry images such as CIFAR [40] and TinyImageNet [41]. With pre-trained models such as CLIP, OOD detection can be extended to more realistic and complex datasets. In this work, we scale up evaluations in terms of (1) image resolution, (2) dataset variety, and (3) number of classes. We consider the following ID datasets: CUB-200 [80], STANFORD-CARS [39], FOOD-101 [6], OXFORD-PET [57] and variants of IMAGENET [11]. For OOD test datasets, we use the same ones in [32], including subsets of iNaturalist [76], SUN [86], PLACES [96], and TEXTURE [10]. For each OOD dataset, the categories are not overlapping with the ID dataset. We also use subsets of ImageNet-1k for fine-grained analysis. For example, we construct ImageNet-10 that mimics the class distribution of CIFAR-10 but with high-resolution images. For hard OOD evaluation, we curate ImageNet-20, which consists of 20 classes semantically similar to ImageNet-10 (*e.g.*, dog (ID) vs. wolf (OOD)).

**Model.** In our experiments, we adopt CLIP [59] as the target pre-trained model, which is one of the most popular and publicly available vision-language models. Note that our method is not limited to CLIP; it can generally be applicable for contrastive vision-language pre-training models that promote multi-modal feature alignment. Specifically, we mainly use CLIP-B/16, which consists of a ViT-B/16 Transformer as the image encoder and a masked self-attention Transformer [77] as the text encoder. To indicate the input patch size in ViT models, we append "/x" to model names. We prepend -B, -L to indicate `Base` and `Large` versions of the corresponding architecture. For instance, ViT-B/16 implies the `Base` variant with an input patch resolution of $16 \times 16$. We also use CLIP-L/14 which is based on ViT-L/14 as a representative of large models. Unless specified otherwise, the temperature $\tau$ is 1 for all experiments. Details of the datasets, experimental setup, and hyperparameters are provided in Appendix B.

**Metrics.** For evaluation, we use the following metrics: (1) the false positive rate (FPR95) of OOD samples when the true positive rate of in-distribution samples is at 95%, (2) the area under the receiver operating characteristic curve (AUROC), and (3) ID classification accuracy (ID ACC).

Table 1: Zero-shot OOD detection with MCM score based on CLIP-B/16 with various ID datasets.

| | OOD Dataset | | | | | | | | Average | |
| | iNaturalist | | SUN | | Places | | Texture | | | |
| ID Dataset | FPR95↓ | AUROC↑ | FPR95↓ | AUROC↑ | FPR95↓ | AUROC↑ | FPR95↓ | AUROC↑ | FPR95↓ | AUROC↑ |
|---|---|---|---|---|---|---|---|---|---|---|
| **CUB-200** [80] | 9.83 | 98.24 | 4.93 | 99.10 | 6.65 | 98.57 | 6.97 | 98.75 | 7.09 | 98.66 |
| **Stanford-Cars** [39] | 0.05 | 99.77 | 0.02 | 99.95 | 0.24 | 99.89 | 0.02 | 99.96 | 0.08 | 99.89 |
| **Food-101** [6] | 0.64 | 99.78 | 0.90 | 99.75 | 1.86 | 99.58 | 4.04 | 98.62 | 1.86 | 99.43 |
| **Oxford-Pet** [57] | 2.85 | 99.38 | 1.06 | 99.73 | 2.11 | 99.56 | 0.80 | 99.81 | 1.70 | 99.62 |
| **ImageNet-10** | 0.12 | 99.80 | 0.29 | 99.79 | 0.88 | 99.62 | 0.04 | 99.90 | 0.33 | 99.78 |
| **ImageNet-20** | 1.02 | 99.66 | 2.55 | 99.50 | 4.40 | 99.11 | 2.43 | 99.03 | 2.60 | 99.32 |
| **ImageNet-100** | 18.13 | 96.77 | 36.45 | 94.54 | 34.52 | 94.36 | 41.22 | 92.25 | 32.58 | 94.48 |

Table 2: OOD detection performance for ImageNet-1k [11] as ID.

| | OOD Dataset | | | | | | | | Average | |
| | iNaturalist | | SUN | | Places | | Texture | | | |
| Method | FPR95↓ | AUROC↑ | FPR95↓ | AUROC↑ | FPR95↓ | AUROC↑ | FPR95↓ | AUROC↑ | FPR95↓ | AUROC↑ |
|---|---|---|---|---|---|---|---|---|---|---|
| | Requires training (or w. fine-tuning) | | | | | | | | | |
| MOS [32] (BiT) | 9.28 | 98.15 | 40.63 | 92.01 | 49.54 | 89.06 | 60.43 | 81.23 | 39.97 | 90.11 |
| Fort et al. [19] (ViT-B) | 15.07 | 96.64 | 54.12 | 86.37 | 57.99 | 85.24 | 53.32 | 84.77 | 45.12 | 88.25 |
| Fort et al. [19] (ViT-L) | 15.74 | 96.51 | 52.34 | 87.32 | 55.14 | 86.48 | 51.38 | 85.54 | 43.65 | 88.96 |
| Energy [48] (CLIP-B) | 21.59 | 95.99 | 34.28 | 93.15 | 36.64 | 91.82 | 51.18 | 88.09 | 35.92 | 92.26 |
| Energy [48] (CLIP-L) | 10.62 | 97.52 | 30.46 | 93.83 | 32.25 | 93.01 | 44.35 | 89.64 | 29.42 | 93.50 |
| MSP [25] (CLIP-B) | 40.89 | 88.63 | 65.81 | 81.24 | 67.90 | 80.14 | 64.96 | 78.16 | 59.89 | 82.04 |
| MSP [25] (CLIP-L) | 34.54 | 92.62 | 61.18 | 83.68 | 59.86 | 84.10 | 59.27 | 82.31 | 53.71 | 85.68 |
| | Zero-shot (no training required) | | | | | | | | | |
| **MCM** (CLIP-B) | 30.91 | 94.61 | 37.59 | 92.57 | 44.69 | 89.77 | 57.77 | 86.11 | 42.74 | 90.77 |
| **MCM** (CLIP-L) | 28.38 | 94.95 | 29.00 | 94.14 | 35.42 | 92.00 | 59.88 | 84.88 | 38.17 | 91.49 |

## 4.2 Main Results

**MCM supports a diverse collection of tasks while being zero-shot.** We first show that zero-shot OOD detection with MCM is effective across a wide variety of tasks—with just *one single pre-trained model*. To showcase the versatility of MCM, we consider the seven ID datasets here. To the best of our knowledge, this is among the first attempts to showcase the efficacy under an expansive and diverse collection of ID datasets. The zero-shot OOD detection performance is summarized in Table 1. A salient observation is that MCM can achieve superior detection performance on many tasks. For example, using STANFORD-CARS as ID, MCM yields an average FPR95 of **0.08**%. Considering that there are no training samples or OOD information involved, these results are very encouraging.

It can be also seen from Table 1 that MCM is promising, especially when the number of samples per ID class is limited in the training set. For example, there are only around 40 samples per class for Stanford-Cars, 100 for Oxford-Pet, and 30 for CUB-200. The sample scarcity makes OOD detection methods that rely on fine-tuning difficult. For example, after fine-tuning on Food-101, while the ID accuracy is increased from $86.3\%$ to $92.5\%$ ↑, OOD detection based on MSP is on par with MCM ($99.5\%$ vs. $99.4\%$ in AUROC).

**MCM scales effectively to large datasets.** To examine the scalability of MCM, we compare it with recent competitive OOD detection methods [19, 32] on the ImageNet-1k dataset (ID) in Table 2. We observe the following trends:

- Larger models lead to superior performance. Compared with CLIP-B, MCM based on CLIP-L reduces FPR95 by $4.57\%$. Zero-shot ID classification accuracy is also improved by $6.27\%$ with the larger model, reaching $73.28\%$ (see Appendix D). This suggests that larger models are endowed with a better representation quality, which benefits both ID classification and OOD detection with MCM. Our finding echos with the recent observations [78] that higher ID classification accuracy is correlated with stronger OOD detection performance.

- MOS [32] recently demonstrated competitive performance on ImageNet-1k, which requires model fine-tuning based on BiT [38]. In contrast, we show that MCM (CLIP-L) outperforms MOS by $1.38\%$ in AUROC while being zero-shot (training-free).

- MCM shares a softmax scaling function with the classic (visual) confidence-based score MSP [25]. To implement MSP, we adopt the commonly used linear probe approach by fine-tuning a linear layer on frozen visual features of CLIP. After fine-tuning, ID accuracy significantly improves, reaching $84.12\%$ (CLIP-L). Interestingly, the OOD detection performance of MSP is worse than

Table 3: Performance comparison on **hard OOD detection** tasks. MCM is competitive on all three hard OOD tasks without training involved. MSP (based on fine-tuned CLIP) does not further improve performance.

| Method | ID OOD | ImageNet-10 ImageNet-20 | ImageNet-20 ImageNet-10 | Waterbirds Spurious OOD |
|---|---|---|---|---|
| | | FPR95 / AUROC | FPR95 / AUROC | FPR95 / AUROC |
| MSP [25] (fine-tuning) | | 9.38 / 98.31 | 12.51 / 97.70 | 39.57 / 90.99 |
| Mahalanobis [42] (visual only) | | 78.32 / 85.60 | 43.03 / 89.94 | 2.21 / 99.55 |
| MCM (zero-shot) | | 5.00 / 98.71 | 12.91 / 98.09 | 5.87 / 98.36 |

MCM by $15.54\%$ in FPR95. Under the same model fine-tuned with linear probing, we observe that the Energy score outperforms MSP, corroborating findings in [48]. We investigate more in Section 5.

- Recently, Fort *et al.* [19] explore small-scale OOD detection by fine-tuning the full ViT model. When extended to large-scale tasks, we find that MCM still yields superior performance under the same image encoder configuration (ViT-B or ViT-L). This further highlights the advantage of utilizing vision-language joint embeddings for large-scale visual OOD detection.

**MCM benefits hard OOD detection.** Going beyond, we investigate whether MCM is still effective for hard OOD inputs. We consider the following two categories of hard OOD:

- **Semantically hard OOD**: OOD samples that are semantically similar to ID samples are particularly challenging for OOD detection algorithms [85]. To evaluate hard OOD detection tasks in realistic settings, here we consider ImageNet-10 (ID) vs. ImageNet-20 (OOD) and vice versa. The pair consists of high-resolution images with semantically similar categories such as dog versus wolf. As shown in Table 3, MCM outperforms Mahalanobis [42] by **73.32%** in FPR95 for ImageNet-10 (ID) vs. ImageNet-20 (OOD) and **30.12%** vice versa.

- **Spurious OOD**: Modern neural networks can exploit spurious correlations for predictions [3]. For example, in the Waterbirds dataset [64], there exist spurious correlations between the habitat (*e.g.*, water) and bird types. A recent work [50] proposes a new type of hard OOD named spurious OOD and shows that most OOD detection methods perform much worse for spurious OOD inputs compared to non-spurious inputs. The spurious OOD inputs are created to share the same background (*i.e.*, water) as ID data but have different object labels (*e.g.*, a boat rather than a bird). See Appendix C for illustrations. The results are shown in Table 3. It has been shown that CLIP representations are robust to distributional shifts [59]. Therefore, while prior works [50] show that spurious OOD inputs are challenging for methods based on ResNet [23], MCM and Mahalanobis scores based on pre-trained CLIP perform much better. On the other hand, fine-tuning exposes the model to the training set containing spurious correlations. As a result, MSP performs much worse than MCM ($39.57\%$ vs. $5.87\%$ in FPR95).

**MCM outperforms CLIP-based baselines.** Two recent works also use CLIP embeddings for OOD detection [16, 19]. However, fundamental limitations exist for both works. Fort *et al.* [19] assume that a candidate OOD label set $\mathcal{Y}_C$ is known, and used $\sum_{y \in \mathcal{Y}_C} \hat{p}(y|\mathbf{x})$ for OOD detection. Here the predictive probability $\hat{p}(y|\mathbf{x})$ is obtained by normalizing the inner products over $|\mathcal{Y}_{\text{in}}| + |\mathcal{Y}_C|$ classes. While applying softmax converts any vector to probabilities, as we show in Section 3, the converted probabilities do not necessarily correspond to $\mathbb{P}(\text{OOD}|\mathbf{x})$. Moreover, obtaining such an OOD label set is typically not feasible, which fundamentally limits its applicability. A recent work [16] realizes this idea by training an extra text decoder on top of CLIP's image encoder to generate candidate labels. However, [16] cannot guarantee the generated labels are non-overlapping with the ID labels.

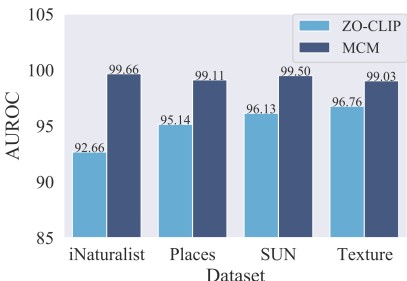

Figure 3: Comparison with a candidate label-based score ZO-CLIP on ImageNet-20, based on our implementation of [16]. Implementation details are deferred to Appendix E.1.

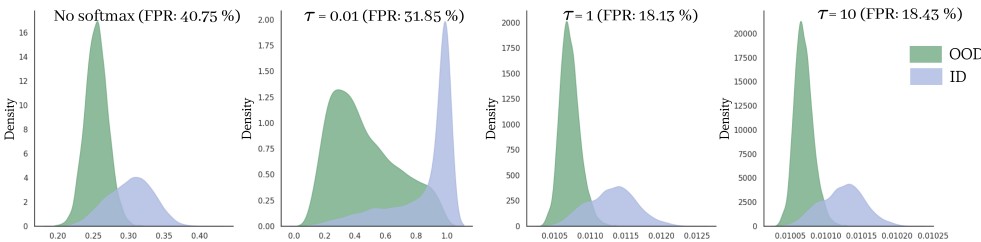

Figure 4: The influence of softmax scaling and temperature. We use ImgeNet-100 (ID) vs. iNaturalist (OOD). Softmax scaling with a moderate temperature significantly improves FPR95.

We enhance the baseline with a stronger decoder and a filter module (see Appendix E.1). As shown in Figure 3, MCM outperforms the enhanced baseline on all OOD datasets. Moreover, MCM is much simpler to use—alleviating the need for an OOD label set or training an additional caption generator. In contrast, the caption generator's performance largely affects OOD detection. Poor caption quality degenerates the OOD detection performance of candidate label-based methods. Moreover, obtaining a reliable caption generator for *any input image* can significantly increase the computational overhead.

## 5    Discussion: A Closer Look at MCM

**Empirical verification on the role of softmax.**    In Section 3, we prove that softmax scaling on cosine similarity scores with a moderate $\tau$ improves the ID-OOD separability. Here we empirically verify our theoretical results. As shown in Figure 4, compared to directly using the maximum cosine similarity without softmax (leftmost figure), softmax scaling with a temperature $\tau = 1$ significantly improves the performance by $22.6\%$ in FPR95, and further increasing $\tau$ (*e.g.*, $\tau = 10$) leads to similar performance. The results are based on ImageNet-100 (ID) versus iNaturalist (OOD).

Now, we verify if our theoretical bound (*c.f.* Theorem 3.1) is satisfied empirically as well in Figure 4. From the leftmost figure, we can estimate $\lambda^{\text{wo}} \approx 0.26$, $\delta \approx 0.03$, and $s_{\hat{y}_2} \approx 0.23$. By checking the third figure ($\tau = 1$ is the temperature value we use for most experiments), we approximate $\lambda \approx 0.011$. As $K = 100$, we plug in the values and obtain the lower bound $T = \frac{\lambda(K-1)\left(\lambda^{\text{wo}}+\delta-s_{\hat{y}_2}\right)}{K\lambda-1} \approx 0.65$. Since $\tau = 1 > 0.65$, by Theorem 3.1, applying softmax scaling with $\tau = 1$ is provably superior to without softmax scaling for OOD detection.

**Are vision-language features better than visual feature alone?**    MCM can be interpreted as a distance-based approach—images that are closer to one of the $K$ class prototypes are more likely to be ID and vice versa. Here the class prototypes are defined based on a textual encoder. Alternatively, one can define the class prototypes based on visual features. For example, Mahalanobis [42] defines a class prototype as the average of visual embeddings for images belonging to the same class. This raises the question whether MCM (with *multi-modal* vision-language features) is better than Mahalanobis (with *single-modal* visual feature). For a fair comparison, we use the same ViT image encoder from CLIP-B. Both MCM and Mahalanobis extract visual features from the penultimate layer. On ImageNet-1k, Mahalanobis displays a limited perfor-

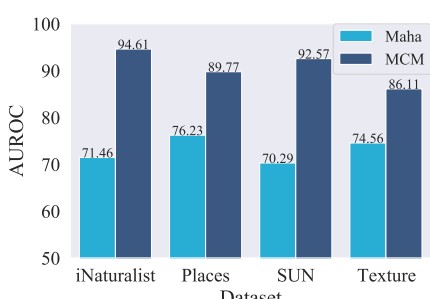

Figure 5: Comparison with Mahalanobis (Maha) score on ImageNet-1k.

mance, with $73.14\%$ AUROC averaged across four OOD test datasets (**90.77%** for MCM), as shown in Figure 5. From a practical perspective, Mahalanobis requires computing the inverse covariance matrix, which can be both computationally expensive and inaccurate when the number of samples is scarce and the number of ID classes grows. In contrast, MCM is easier to use and more robust.

**MCM without softmax scaling.**    In Section 3, we provide theoretical justifications for the necessity of softmax scaling for CLIP-like models. To further verify our observations empirically, we show OOD detection performance based on the maximum cosine similarity score $S_{\text{MCM}}^{\text{wo}}(\mathbf{x}'; \mathcal{Y}_{\text{in}}, \mathcal{T}, \mathcal{I}) = \max_{i \in [K]} s_i(\mathbf{x}')$. The results are shown in Table 4. For easy tasks such as Food-101 [39], Stanford-

Table 4: Zero-shot OOD detection of $S_{\mathrm{MCM}}^{\mathrm{wo}}$ based on CLIP-B/16.

| ID Dataset | OOD Dataset | | | | | | | | Average | |
| | iNaturalist | | SUN | | Places | | Texture | | | |
| | FPR95↓ | AUROC↑ | FPR95↓ | AUROC↑ | FPR95↓ | AUROC↑ | FPR95↓ | AUROC↑ | FPR95↓ | AUROC↑ |
|---|---|---|---|---|---|---|---|---|---|---|
| Stanford-Cars [39] | 0.00 | 100 | 0.02 | 99.99 | 0.26 | 99.94 | 0.00 | 100 | 0.07 | 99.98 |
| Food-101 [6] | 0.56 | 99.86 | 0.09 | 99.95 | 0.49 | 99.88 | 8.33 | 97.44 | 2.37 | 99.28 |
| Oxford-Pet [57] | 0.02 | 99.98 | 0.05 | 99.97 | 0.20 | 99.94 | 0.27 | 99.91 | 0.14 | 99.95 |
| ImageNet-10 | 2.40 | 99.42 | 1.79 | 99.55 | 2.83 | 99.32 | 1.86 | 99.56 | 2.22 | 99.46 |
| ImageNet-20 | 14.96 | 97.87 | 13.10 | 97.97 | 14.21 | 97.67 | 13.46 | 97.32 | 13.93 | 97.71 |
| ImageNet-1k | 61.66 | 89.31 | 64.39 | 87.43 | 63.67 | 85.95 | 86.61 | 71.68 | 69.08 | 83.59 |

Cars [39], and Oxford-Pet [57] as ID, the performance of maximum cosine similarity score is similar to MCM (see Table 1 and Table 2). However, for more challenging tasks such as ImageNet-20 and ImageNet-1k, MCM significantly outperforms that without softmax scaling. For example, the average FPR95 is improved by **11.33%** on ImageNet-20 and **26.34%** on ImageNet-1k, which highlights the necessity of a proper scaling function for CLIP-based OOD detection.

**MCM for ResNet-based CLIP models.** Our main results are based on the CLIP model with ViT image encoder. We additionally investigate the effectiveness of MCM on ResNet-based CLIP. Specifically, we use RN50x4 (178.3M), which shares a similar number of parameters as CLIP-B/16 (149.6M). The results are shown in Table 5. We can see that MCM still shows promising results with ResNet-based CLIP models, and the performance is comparable between RN50x4 and CLIP-B/16 (89.97 vs. 90.77 in AUROC).

Table 5: Comparison with ResNet-based CLIP models on ImageNet-1k (ID).

| Model | OOD Dataset | | | | | | | | Average | |
| | iNaturalist | | SUN | | Places | | Texture | | | |
| | FPR95↓ | AUROC↑ | FPR95↓ | AUROC↑ | FPR95↓ | AUROC↑ | FPR95↓ | AUROC↑ | FPR95↓ | AUROC↑ |
|---|---|---|---|---|---|---|---|---|---|---|
| RN50x4 | 44.51 | 91.51 | 35.11 | 92.84 | 43.74 | 89.60 | 57.73 | 85.93 | 45.27 | 89.97 |
| CLIP-B/16 | 30.91 | 94.61 | 37.59 | 92.57 | 44.69 | 89.77 | 57.77 | 86.11 | 42.74 | 90.77 |

**Effect of prompt ensembling.** We examine MCM's performance with prompt ensembling. For example, Radford *et al.* [59] create 80 possible prompts according to the image modalities and nuances in ImageNet. We experiment with the two prompt sets, one of size 80 as in [59], and our own set of 5 prompts. Ensembles are obtained by averaging the textual features. As expected, using ensembles increases the ID classification accuracy on ImageNet-1k (2% with CLIP-B and 3% with CLIP-L). For OOD detection, the average FPR95 is reduced from 38.17% with the default prompt to 35.23%↓ with an ensemble of five prompts shown in Table 6. In addition, the detection performance with 5 prompts is slightly better than with 80 prompts. Note that prompt ensembling does not increase the inference-time cost, as the textual embeddings (across many prompts) can be pre-calculated and averaged into a single embedding.

A photo of a <label>.
A blurry photo of a <label>.
A photo of many <label>.
A photo of the large <label>.
A photo of the small <label>.

Table 6: The five prompt templates.

## 6 Related Works

**OOD detection in computer vision.** For open-world multi-class classification, the goal of OOD detection is to derive a binary ID-OOD classifier along with a multi-class classification model for visual inputs. A plethora of methods has been proposed for deep neural networks [91], including generative model-based methods [7, 20, 36, 53, 54, 56, 61, 67, 88], and discriminative-model based methods. For the latter category, an OOD score can be derived based on the softmax output [4, 12, 24, 25, 29, 32, 46, 90], energy-based score [15, 48, 49, 69, 70, 82], gradient information [31], or the feature embeddings [14, 42, 65, 66, 71, 72, 85] of a model. Morteza *et al.* [52], Fang *et al.* [17], and Bitterwolf *et al.* [5] provided theoretical analysis for OOD detection. Recent works [63, 83] also explored OOD detection for long-tailed distributions. Works insofar have mostly focused on OOD detection for a task-specific model using only visual information. In contrast, we explore a novel

paradigm of zero-shot OOD detection that incorporates rich textual information and can perform a wide variety of tasks.

**OOD detection in natural language processing.** Distribution shifts can occur due to the change of topics and domains, unexpected user utterances, *etc*. Challenging benchmarks [37] and characterization of distributional shifts [1] have been proposed in recent years. Compared to early language models such as ConvNets and LSTM [28], pre-trained language models are more robust to distribution shifts and more effective at identifying OOD instances [26, 58, 89]. Various algorithmic solutions are proposed to handle OOD detection, including outlier exposure [30], model ensembling [44], data augmentation [8, 93, 95], contrastive learning [34, 98], and an auxiliary module that incorporates domain labels [68]. Tan *et al.* [73] also explore zero-shot OOD detection for text classification tasks. However, prior works focus on pure natural language processing (NLP) settings, while we explore utilizing textual embeddings for zero-shot *visual* OOD detection.

**Vision-language models.** Utilizing large-scale pre-trained vision-language models for multimodal downstream tasks has become an emerging paradigm with remarkable performance [22, 74]. In general, two types of architectures exist: single-stream models like VisualBERT [43] and ViLT [35] feed the concatenated text and visual features into a single transformer-based encoder; dual-stream models such as CLIP [59], ALIGN [33], and FILIP [92] use separate encoders for text and image and optimize with contrastive objectives to align semantically similar features in different modalities. In particular, CLIP enjoys popularity due to its simplicity and strong performance. CLIP-like models inspire numerous follow-up works [45, 94, 97], which aim to improve data efficiency and better adaptation to downstream tasks. This paper adopts CLIP as the target pre-trained model, but our approach can be generally applicable to contrastive models that promote vision-language alignment.

**Multi-modal OOD detection.** Exploring textual information for visual OOD detection is a new area with limited existing works. Fort *et al.* [19] propose to feed the potential OOD labels to the textual encoder of CLIP [59]. Recently, Esmaeilpour *et al.* [16] propose to train a label generator based on the visual encoder of CLIP and use the generated labels for OOD detection. While both works rely on a set of candidate OOD labels, MCM is OOD-agnostic and alleviates the need for prior information on OOD. Moreover, prior works [16, 59] only focus on small-scale inputs. We largely expand the scope to a wide range of large-scale realistic datasets, and show new theoretical insights.

## 7 Conclusion

In this work, we delve into a new landscape for OOD detection, departing from the classic single-modal toward a multi-modal regime. By viewing the textual features as the "concept prototypes", we explore a new OOD detection approach MCM, based on the joint vision-language representations. Unlike the majority of OOD detection methods, MCM offers several compelling advantages: training-free, generalizable to many tasks, scalable to hundreds of classes, and does not require any prior information on OOD inputs. Moreover, we provide theoretical guarantees on how softmax scaling provably improves zero-shot OOD detection. We investigate the effectiveness of MCM on a wide range of large-scale realistic tasks, including several types of hard OOD datasets. Lastly, we demonstrate the advantage of vision-language features over pure visual features for OOD detection. We hope our work will inspire future research toward multi-modal OOD detection.

## Acknowledgement

The authors wish to thank Junjie Hu, Ying Fan, Ruisu Zhang, Andrew Geng, and Soumya Suvra Ghosal for the helpful discussions. The work is supported by a Google-Initiated Research Grant, and gift funding from Adobe Research.

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
