# A Theoretical Justification: Softmax Scaling for Zero-Shot OOD Detection

In this section, we provide the proof for Theorem 3.1 in Section 3, which states the benefits of applying softmax scaling to inner products for OOD detection. We begin with a review of notations.

**Notations.** We denote the text encoder of a pre-trained CLIP-like model as $\mathcal{T} : t \to \mathbb{R}^d$ and the image encoder $\mathcal{I} : \mathbf{x} \to \mathbb{R}^d$. For a given task with label set $\mathcal{Y}_{\text{in}} = \{y_1, y_2, ..., y_K\}$, we construct a collection of concept vectors $\mathcal{T}(t_i)$. For a given input $\mathbf{x}'$, we denote the cosine similarity *w.r.t.* concept vectors as $s_i(\mathbf{x}') = \frac{\mathcal{I}(\mathbf{x}') \cdot \mathcal{T}(t_i)}{\|\mathcal{I}(\mathbf{x}')\| \cdot \|\mathcal{T}(t_i)\|} \forall i \in [K]$, where $|s_i(\mathbf{x}')| \leq B$ for all $\mathbf{x}' \in \mathcal{X}$.[2] We define the maximum concept matching (MCM) score as: $S_{\text{MCM}}(\mathbf{x}'; \mathcal{Y}_{\text{in}}, \mathcal{T}, \mathcal{I}) = \max_{i \in [K]} \frac{e^{s_i(\mathbf{x}')/\tau}}{\sum_{j=1}^{K} e^{s_j(\mathbf{x}')/\tau}}$.
We denote the maximum inner product without applying softmax scaling as $S_{\text{MCM}}^{\text{wo}}(\mathbf{x}'; \mathcal{Y}_{\text{in}}, \mathcal{T}, \mathcal{I}) = \max_{i \in [K]} s_i(\mathbf{x}')$. By convention, the OOD detection functions are given by:

$$G^{\text{wo}}(\mathbf{x}'; \mathcal{Y}_{\text{in}}, \mathcal{T}, \mathcal{I}) = \begin{cases} 1 & S_{\text{MCM}}^{\text{wo}}(\mathbf{x}'; \mathcal{Y}_{\text{in}}, \mathcal{T}, \mathcal{I}) \geq \lambda^{\text{wo}} \\ 0 & S_{\text{MCM}}^{\text{wo}}(\mathbf{x}'; \mathcal{Y}_{\text{in}}, \mathcal{T}, \mathcal{I}) < \lambda^{\text{wo}} \end{cases},$$

$$G(\mathbf{x}'; \mathcal{Y}_{\text{in}}, \mathcal{T}, \mathcal{I}) = \begin{cases} 1 & S_{\text{MCM}}(\mathbf{x}'; \mathcal{Y}_{\text{in}}, \mathcal{T}, \mathcal{I}) \geq \lambda \\ 0 & S_{\text{MCM}}(\mathbf{x}'; \mathcal{Y}_{\text{in}}, \mathcal{T}, \mathcal{I}) < \lambda \end{cases},$$

**Remarks:** By convention, 1 represents the positive class (ID) and 0 indicates OOD; $\lambda$ and $\lambda^{\text{wo}}$ are typically chosen such that the true positive rate is at 95%.

For convenience, we paste the assumptions and the theorem in Section 3 below,

**Assumption A.1.** Let $z := \mathbb{1}\{y \in \mathcal{Y}_{\text{in}}\}$ and $Q_{\mathbf{x}}$ denotes the out-of-distribution $\mathbb{P}_{\mathbf{x}|z=0}$ (marginal distribution of $\mathbf{x}$ conditioned on $z = 0$). Assume $\exists \delta > 0$ such that

$$Q_{\mathbf{x}} \left( \frac{1}{K-1} \sum_{i \neq \hat{y}} [s_{\hat{y}_2}(\mathbf{x}) - s_i(\mathbf{x})] < \delta \right) = 1,$$

where $\hat{y} := \text{argmax}_{i \in [K]} s_i(\mathbf{x})$ and $\hat{y}_2 := \text{argmax}_{i \neq \hat{y}, i \in [K]} s_i(\mathbf{x})$ denote the indices of the largest and second largest cosine similarities for an OOD input $\mathbf{x}$.

**Theorem A.1.** Given a pre-trained CLIP-like model $(\mathcal{T}, \mathcal{I})$ and a task with label set $\mathcal{Y}_{\text{in}} = \{y_1, y_2, ..., y_K\}$. If $Q_{\mathbf{x}}$ satisfy Assumption A.1, Then there exists a constant $T = \frac{\lambda(K-1)(\lambda^{\text{wo}}+\delta-s_{\hat{y}_2})}{K\lambda-1}$ such that for any temperature $\tau > T$, we have:

$$\text{FPR}(\tau, \lambda) \leq \text{FPR}^{\text{wo}}(\lambda^{\text{wo}}),$$

where $\text{FPR}(\tau, \lambda)$ is the false positive rate based on softmax scaling with temperature $\tau$ and threshold $\lambda$; $\text{FPR}^{\text{wo}}(\lambda^{\text{wo}})$ is the false positive rate without softmax scaling based on threshold $\lambda^{\text{wo}}$. This suggests that applying softmax scaling with temperature results in superior OOD detection performance compared to without softmax scaling.

*Proof.* By definition, we express the false positive rate $\text{FPR}(\tau, \lambda)$ as follows,

$$\begin{aligned} \text{FPR}(\tau, \lambda) &= \mathbb{P}\left(G(\mathbf{x}'; \mathcal{Y}_{\text{in}}, \mathcal{T}, \mathcal{I}) = 1 \mid z = 0\right) \\ &= Q_{\mathbf{x}'}\left(G(\mathbf{x}'; \mathcal{Y}_{\text{in}}, \mathcal{T}, \mathcal{I}) = 1\right) \\ &= Q_{\mathbf{x}'}\left(p_{\hat{y}}(\mathbf{x}'; \tau) > \lambda\right) \\ &= Q_{\mathbf{x}'}\left(\frac{e^{s_{\hat{y}}(\mathbf{x}')/\tau}}{\sum_{j=1}^{K} e^{s_j(\mathbf{x}')/\tau}} > \lambda\right) \\ &= Q_{\mathbf{x}'}\left(\frac{1}{\lambda} > \sum_{i=1}^{K} \exp\left(\frac{s_i(\mathbf{x}') - s_{\hat{y}}(\mathbf{x}')}{\tau}\right)\right) \end{aligned}$$

[2]In practice, we observe that $s_i \in [0.1, 0.3]$ for CLIP with high probability.

By inequality $e^x \geq 1 + x$, we have,

$$Q_{\mathbf{x}'}\left(\frac{1}{\lambda} > \sum_{i=1}^{K} \exp\left(\frac{s_i(\mathbf{x}') - s_{\hat{y}}(\mathbf{x}')}{\tau}\right)\right) \leq Q_{\mathbf{x}'}\left(\frac{1}{\lambda} > \sum_{i=1}^{K}\left[1 + \frac{s_i(\mathbf{x}') - s_{\hat{y}}(\mathbf{x}')}{\tau}\right]\right)$$

This indicates

$$Q_{\mathbf{x}'}\left(\frac{1}{\lambda} > \sum_{i=1}^{K} \exp\left(\frac{s_i(\mathbf{x}') - s_{\hat{y}}(\mathbf{x}')}{\tau}\right)\right) \leq Q_{\mathbf{x}'}\left(\frac{1}{\lambda} > \sum_{i=1}^{K}\left[1 + \frac{s_i(\mathbf{x}') - s_{\hat{y}}(\mathbf{x}')}{\tau}\right]\right)$$

$$= Q_{\mathbf{x}'}\left(\sum_{i=1}^{K}(s_{\hat{y}}(\mathbf{x}') - s_i(\mathbf{x}')) > \left(K - \frac{1}{\lambda}\right)\tau\right)$$

Since

$$\sum_{i=1}^{K}(s_{\hat{y}}(\mathbf{x}') - s_i(\mathbf{x}')) = \sum_{i \neq \hat{y}}(s_{\hat{y}}(\mathbf{x}') - s_{\hat{y}_2}(\mathbf{x}') + s_{\hat{y}_2}(\mathbf{x}') - s_i(\mathbf{x}'))$$

$$= \sum_{i \neq \hat{y}}(s_{\hat{y}}(\mathbf{x}') - s_{\hat{y}_2}(\mathbf{x}')) + \sum_{i \neq \hat{y}}(s_{\hat{y}_2}(\mathbf{x}') - s_i(\mathbf{x}'))$$

$$= (K-1)(s_{\hat{y}}(\mathbf{x}') - s_{\hat{y}_2}(\mathbf{x}')) + \sum_{i \neq \hat{y}}(s_{\hat{y}_2}(\mathbf{x}') - s_i(\mathbf{x}'))$$

By Assumption 3.1, we have

$$Q_{\mathbf{x}'}\left(\sum_{i=1}^{K}(s_{\hat{y}}(\mathbf{x}') - s_i(\mathbf{x}')) < (K-1)(s_{\hat{y}}(\mathbf{x}') - s_{\hat{y}_2}(\mathbf{x}')) + (K-1)\delta\right) = 1.$$

Therefore,

$$Q_{\mathbf{x}'}\left(\sum_{i=1}^{K}(s_{\hat{y}}(\mathbf{x}') - s_i(\mathbf{x}')) > \left(K - \frac{1}{\lambda}\right)\tau\right) \leq Q_{\mathbf{x}'}\left(s_{\hat{y}}(\mathbf{x}') - s_{\hat{y}_2}(\mathbf{x}') > -\delta_2 + \left(K - \frac{1}{\lambda}\right)\frac{\tau}{K-1}\right)$$

$$= Q_{\mathbf{x}'}\left(s_{\hat{y}}(\mathbf{x}') - s_{\hat{y}_2}(\mathbf{x}') > -\delta_2 + \lambda'\right)$$

$$= Q_{\mathbf{x}'}\left(s_{\hat{y}}(\mathbf{x}') > s_{\hat{y}_2}(\mathbf{x}') - \delta_2 + \lambda'\right),$$

where $\lambda' = \left(K - \frac{1}{\lambda}\right)\frac{\tau}{K-1}$ is a monotonic function of $\lambda$ (*i.e.*, minimizing false positive rate *w.r.t.* $\lambda$ is equivalent to minimizing *w.r.t.* $\lambda'$.)

For $\tau > 0$, we can rewrite the MCM score as

$$S_{\text{MCM}}(\mathbf{x}'; \mathcal{Y}_{\text{in}}, \mathcal{T}, \mathcal{I}) = \max_{i \in [K]} \frac{e^{s_i(\mathbf{x}')/\tau}}{\sum_{j=1}^{K} e^{s_j(\mathbf{x}')/\tau}} = \frac{e^{s_{\hat{y}}(\mathbf{x}')/\tau}}{\sum_{j=1}^{K} e^{s_j(\mathbf{x}')/\tau}}$$

$$= \frac{1}{1 + \sum_{j=1, j \neq \hat{y}}^{K} e^{(s_j(\mathbf{x}') - s_{\hat{y}}(\mathbf{x}'))/\tau}}$$

As $\hat{y} := \text{argmax}_{i \in [K]} s_i(\mathbf{x})$, $s_j(\mathbf{x}') - s_{\hat{y}}(\mathbf{x}') \leq 0$, $S_{\text{MCM}}(\mathbf{x}'; \mathcal{Y}_{\text{in}}, \mathcal{T}, \mathcal{I})$ is a monotonically decreasing function of $\tau$, we have:

$$S_{\text{MCM}}(\mathbf{x}'; \mathcal{Y}_{\text{in}}, \mathcal{T}, \mathcal{I}) > \lim_{\tau \to \infty} \frac{1}{1 + \sum_{j=1, j \neq \hat{y}}^{K} e^{(s_j(\mathbf{x}') - s_{\hat{y}}(\mathbf{x}'))/\tau}} = \frac{1}{K}$$

Therefore by the definition of $\lambda$, we have $\lambda > \frac{1}{K}$, $\lambda' = \left(K - \frac{1}{\lambda}\right)\frac{\tau}{K-1} > 0$

For moderately large $\tau > T$ where $T = \frac{\lambda(K-1)(\lambda^{\text{wo}} + \delta - s_{\hat{y}_2})}{K\lambda - 1}$, we always have $s_{\hat{y}_2}(\mathbf{x}') - \delta + \lambda' > \lambda^{\text{wo}}$. Therefore, we obtain the following inequality,

$$\text{FPR}(\tau, \lambda) \leq Q_{\mathbf{x}'}\left(s_{\hat{y}}(\mathbf{x}') > s_{\hat{y}_2}(\mathbf{x}') - \delta_2 + \lambda'\right) \leq Q_{\mathbf{x}'}\left(s_{\hat{y}}(\mathbf{x}') > \lambda^{\text{wo}}\right) := \text{FPR}^{\text{wo}}(\lambda^{\text{wo}}),$$

which means that the FPR without softmax scaling is larger than that with softmax scaling and a moderately large temperature. We show in Section 5 that the bound is indeed satisfied in practice with a large-scale ID dataset. $\qquad\square$

# B  Experimental Details

## B.1  Software and Hardware

All methods are implemented in Pytorch 1.10. We run all OOD detection experiments on NVIDIA GeForce RTX-2080Ti GPU and use NVIDIA A100 GPU for fine-tuning CLIP and ViT.

## B.2  Hyperparameters

The only hyperparameter in MCM is the (temperature) scaling factor $\tau$. We use $\tau = 1$ by default unless otherwise specified. Our experiments suggest that MCM is insensitive to the scaling factor, where $\tau$ in a wide range of $[0.5, 100]$ shares similar performance.

## B.3  Datasets

**ImageNet-10** We create ImageNet-10 that mimics the class distribution of CIFAR-10 but with high-resolution images. It contains the following categories (with class ID): warplane (n04552348), sports car (n04285008), brambling bird, (n01530575), Siamese cat (n02123597), antelope (n02422699), Swiss mountain dog (n02107574), bull frog (n01641577), garbage truck (n03417042), horse (n02389026), container ship (n03095699).

**ImageNet-20** For hard OOD evaluation with realistic datasets, we curate ImageNet-20, which consists of 20 classes semantically similar to ImageNet-10 (*e.g.*, dog (ID) vs. wolf (OOD)). The categories are selected based on the distance in the WordNet synsets [18]. Specifically, it contains the following categories: sailboat (n04147183), canoe (n02951358), balloon (n02782093), tank (n04389033), missile (n03773504), bullet train (n02917067), starfish (n02317335), spotted sala-mander (n01632458), common newt (n01630670), zebra (n01631663), frilled lizard (n02391049), green lizard (n01693334), African crocodile (n01697457), Arctic fox (n02120079), timber wolf (n02114367), brown bear (n02132136), moped (n03785016), steam locomotive (n04310018), space shuttle (n04266014), snowmobile (n04252077).

We hope the above two datasets will help future research on large-scale hard OOD detection. We provide a script for generating the datasets at `https://github.com/deeplearning-wisc/MCM`.

**ImageNet-100** We randomly sample 100 classes from ImageNet-1k to curate ImageNet-100. To facilitate reproducibility, the script for generating the dataset and the class list are provided at `https://github.com/deeplearning-wisc/MCM`.

**Conventional (non-spurious) OOD datasets**  Huang *et al.* [32] curate a diverse collection of subsets from iNaturalist [76], SUN [86], Places [96], and Texture [10] as large-scale OOD datasets for ImageNet-1k, where the classes of the test sets do not overlap with ImageNet-1k. We provide a brief introduction to each dataset as follows.

**iNaturalist** contains images in the natural world [76]. It has 13 super-categories and 5,089 sub-categories covering plants, insects, birds, mammals, and so on. We use the subset that contains 110 plant classes not overlapping with ImageNet-1k.

**SUN** stands for the Scene UNderstanding Dataset [86]. SUN contains 899 categories that cover more than indoor, urban, and natural places with or without human beings appearing. We use the subset which contains 50 natural objects not showing in ImageNet-1k.

**Places** is a large scene photographs dataset [96]. It contains photos that are labeled with scene semantic categories from three macro-classes: Indoor, Nature, and Urban. The subset we use is sampled from 50 categories that are not present in ImageNet-1k.

**Texture** stands for the Describable Textures Dataset [10]. It contains images of textures and abstracted patterns. As no categories overlap with ImageNet-1k, we use the entire dataset as in [32].

## B.4  Baselines and sources of model checkpoints

For the Mahalanobis score [42], we use the feature embeddings without $l_2$ normalization as Gaussian distributions naturally do not fit hyperspherical features. Alternatively, one can normalize the embeddings first and then apply the Mahalanobis score.

For Fort *et al.* [19] in Table 2, we fine-tune the whole ViT model on the ID dataset. Specifically, we use the publicly available checkpoints from Hugging Face where the ViT model is pre-trained on ImageNet-21k and fine-tuned on ImageNet-1k. For example, the checkpoint for ViT-B is available at `https://huggingface.co/google/vit-base-patch16-224`.

For CLIP models, our reported results are based on checkpoints provided by Hugging Face for CLIP-B `https://huggingface.co/openai/clip-vit-base-patch16` and CLIP-L `https://huggingface.co/openai/clip-vit-large-patch14`. Similar results can be obtained with checkpoints in the codebase by OpenAI `https://github.com/openai/CLIP`. Note that for CLIP (RN50x4), which is not available in Hugging Face, we use the checkpoint provided by OpenAI.

## C Spurious OOD Datasets

In general, spurious attributes refer to statistically informative features that co-exist with the majority of ID samples but do not necessarily capture cues related to the labels such as color, texture, background, etc [2, 3, 21, 87, 99]. A recent work [50] investigated a new type of hard OOD samples (called spurious OOD) that contain spurious or environmental features, but no object features related to the ID classes. A concrete example is shown in Figure 6, where images of birds co-occur frequently with either the land background or water background. Modern neural networks can spuriously rely on the image background (*e.g.*, water or land) for classification instead of learning to recognize the actual object [62]. Ming *et al.* [50] show that spurious OOD samples remain challenging for most common OOD detection methods based on pure vision models such as ResNet [23].

For ID dataset, we use Waterbirds [64], which combines bird photographs from CUB-200 [80] with water or land background images from PLACES [96]. For the spurious OOD dataset, we use the one created in [50] consisting of land and water background from Places [96].

ID Samples       Spurious OOD

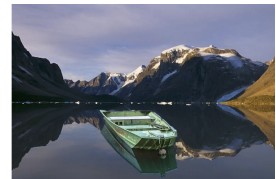

type: landbird     type: waterbird     type: other obj
env: land          env: water         env: water

Figure 6: Illustration of spurious OOD samples for Waterbirds [64]. Images are taken from [50].

## D ID Classification Accuracy

Table 7 shows the multi-class classification accuracy on ImageNet-1k for methods in Table 2.

Table 7: ID classification accuracy on ImageNet-1k (%)

| Method | ID ACC |
|---|---|
| **zero-shot** | |
| MCM (CLIP-B/16) | 67.01 |
| MCM (CLIP-L/14) | 73.28 |
| **w. fine-tuning** | |
| MSP (CLIP-B/16) | 79.39 |
| MSP (CLIP-L/14) | 84.12 |
| Energy [48] (CLIP-B/16) | 79.39 |
| Energy [48] (CLIP-L/14) | 84.12 |
| Fort et al. [19] (ViT-B/16) | 81.25 |
| Fort et al. [19] (ViT-L/14) | 84.05 |
| MOS [32] (BiT) | 75.16 |

# E Implementation of CLIP-Based Baselines

## E.1 Overview of Baselines

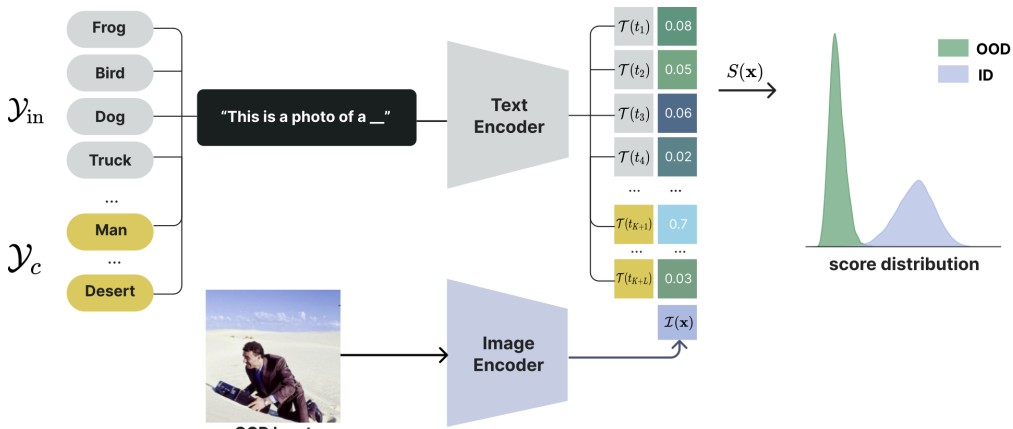

Figure 7: Zero-shot OOD detection with candidate OOD labels. The ID classification task is defined by a set of class labels $\mathcal{Y}_{\text{in}}$. With an additional set of candidate labels $\mathcal{Y}_{\text{C}}$ that describes the contents of the input image, the OOD detection scoring function can be calculated by normalizing over the expanded space of cosine similarities.

We review two previous works on CLIP-based OOD detection [16, 19] in Figure 7, which derive the scoring function based on candidate OOD labels. For a given task with ID label set $\mathcal{Y}_{\text{in}} = \{y_1, y_2, ..., y_K\}$ and candidate labels $\mathcal{Y}_{\text{C}} = \{y_{K+1}, y_{K+2}, ..., y_{K+L}\}$, where ideally $\mathcal{Y}_{\text{in}} \cap \mathcal{Y}_{\text{C}} = \emptyset$, they construct a collection of text embeddings $\mathcal{T}(t_i), i \in \{1, 2, ..., K+L\}$. Here, $t_i$ is the text prompt "`this is a photo of a` $\langle y_i \rangle$" for a label $y_i$. For any test input image $\mathbf{x}$, we can calculate the label-wise matching score based on the cosine similarity between the image and text features: $s_i(\mathbf{x}) = \frac{\mathcal{I}(\mathbf{x}) \cdot \mathcal{T}(t_i)}{\|\mathcal{I}(\mathbf{x}')\| \cdot \|\mathcal{T}(t_i)\|}$. Therefore, a detection score can be derived as:

$$S(\mathbf{x}; \mathcal{Y}_{\text{in}}, \mathcal{Y}_{\text{C}}, \mathcal{T}, \mathcal{I}) = \sum_{i=1}^{K} \frac{e^{s_i(\mathbf{x})/\tau}}{\sum_{j=1}^{K+L} e^{s_j(\mathbf{x})/\tau}},$$

where $\tau > 0$ is the temperature scaling hyperparameter.

## E.2 Obtaining OOD Candidate Labels

For the baseline methods, obtaining OOD candidate labels is a major challenge and limitation. Recently, [16] propose ZO-CLIP, where a transformer (decoder) based on the image encoder of CLIP is used to generate candidate labels. The transformer is trained from scratch on the COCO dataset [47] with simple teacher forcing algorithms. Although the decoder trained on COCO may work well on CIFAR (ID), it does not scale up to large-scale datasets such as ImageNet [11] where categories are not covered in COCO. As a result, [16] only test on small-scale datasets with common classes such as CIFAR (ID).

We improve the baseline by using a high-quality caption generator pre-trained on much larger datasets, which not only saves computational overhead but can potentially improve the quality of generated labels. The pipeline involves three components (see Figure 8):

- A caption generator. Given an input image, it generates a caption serving as the textual description of the input. In this work, we consider ClipCap [51], which uses GPT-2 [60] to generate captions based on CLIP's image encoder. ClipCap is pre-trained on a much larger dataset Conceptual Captions [55] compared to COCO, which can be viewed as an enhanced version of the ZO-CLIP baseline [16]. The checkpoints are publicly available[3].

- A syntactic parser. Given a caption, we extract noun objects using a parsing toolkit released by spaCy [4]. Those nouns can be used as candidate labels $\mathcal{Y}_C$ of the input image.

---

[3]`https://github.com/rmokady/CLIP_prefix_caption`
[4]`https://spacy.io/models/en`

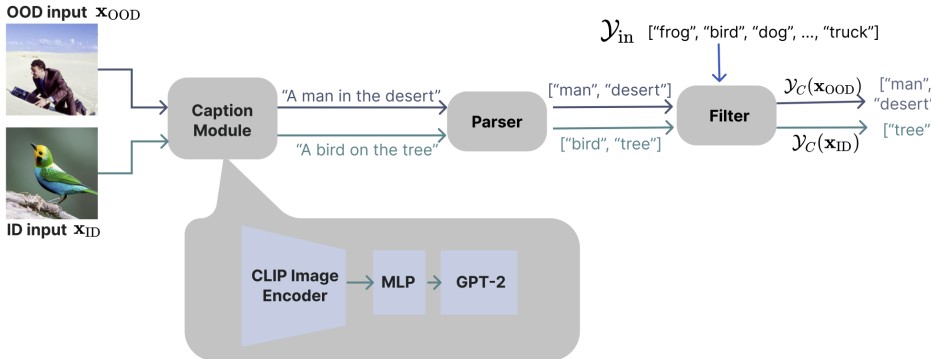

Figure 8: Improved pipeline to generate candidate OOD labels. It consists of three main components: a caption generator, a syntactic parser, and a filtering module to remove candidate labels that overlap with the ID label set.

- A filter module. Unlike [16], we further enhance the baseline by adopting a filtering technique to remove overlapping categories in $\mathcal{Y}_C$ with ID labels $\mathcal{Y}_{in}$, which we detail below.

### E.3  Label Filtering

**Example.** To illustrate the effects of filtering, we begin with a concrete example where ID labels are ["frog","bird"... "truck"], as shown in Figure 8. The generated labels (without filtering) of an ID input of a bird sitting on a tree are ["bird", "tree"]. Therefore, $\mathcal{Y}_{in} \cup \mathcal{Y}_C =$["frog","bird"... "truck","bird", "tree"]. Ideally, the softmax probability distribution over the concatenated labels would be $[0, 0.5, 0, \ldots, 0.5, 0]$ and by definition $S(\mathbf{x}) \approx 0.5$. However, if we filter the generated labels to eliminate nouns with similar meanings as ID, our concatenated labels would be ["frog","bird"... "truck","tree"] and the probability vector would be $[0, 1, 0, \ldots, 0]$, which leads to a much higher score $S(\mathbf{x}) = 1$. In contrast, the generated labels for an OOD input with a caption "man in the desert" would be ["man", "desert"]. The resulting probability vector would be $[0, 0, 0, \ldots, 1, 0]$ and the score $S(\mathbf{x}) = 0$. Therefore, filtering makes it easier to separate ID inputs from OOD inputs (*c.f.* Figure 9).

**String-based filtering.** To implement the idea of filtering, we need a measurement of the similarity between the generated labels and ID labels. The simplest way is string-based filtering where a generated label is filtered if it matches any ID labels (in the string format), as in the case above.

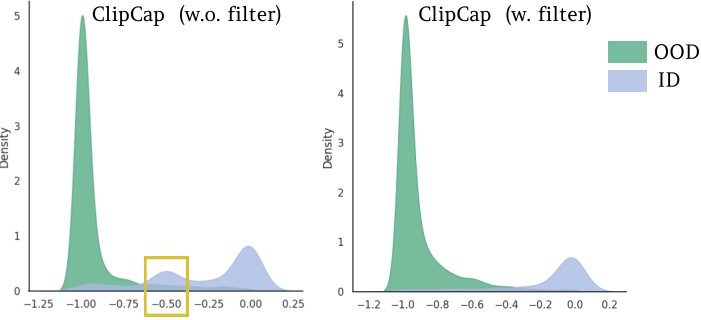

Figure 9: Score distributions for ImageNet-10 (ID) and iNaturalist (OOD) inputs. Simple string-based filtering alleviates the overlap between OOD inputs and ID inputs especially with scores around 0.5 (yellow rectangle), resulting in better ID-OOD separability.

## F  Alternative Scoring Functions

We explore the effectiveness of several alternative scoring functions:

- Entropy: the (negative) entropy of softmax scaled cosine similarities denoted as $S_{\text{entropy}}$;
- Var: the variance of the cosine similarities denoted as $S_{\text{var}}$;
- Scaled: the scaled difference between the largest and second-largest cosine similarities $S_{\text{scaled}} := e^{s_{\hat{y}}(\mathbf{x}) - s_{\hat{y}_2}(\mathbf{x})}$ where $\hat{y} := \text{argmax}_{i \in [K]} s_i(\mathbf{x})$ and $\hat{y}_2 := \text{argmax}_{i \neq \hat{y}, i \in [K]} s_i(\mathbf{x})$.

As shown in Table 8, MCM still gives the most promising results compared to the other three alternative scores across most OOD test sets.

Table 8: Comparison with other scaling functions (applied to inner products) on the large-scale benchmark ImageNet-1k (ID). We use CLIP-B/16 as the backbone.

| Method | OOD Dataset | | | | | | | | Average | |
| | iNaturalist | | SUN | | Places | | Texture | | | |
| | FPR↓ | AUROC↑ | FPR↓ | AUROC↑ | FPR↓ | AUROC↑ | FPR↓ | AUROC↑ | FPR↓ | AUROC↑ |
|---|---|---|---|---|---|---|---|---|---|---|
| Entropy | 84.44 | 63.50 | 93.79 | 62.54 | 94.10 | 64.15 | 97.16 | 58.98 | 92.37 | 62.29 |
| Var | 87.42 | 63.87 | 68.71 | 81.02 | 76.28 | 75.38 | 80.04 | 71.90 | 78.11 | 73.04 |
| Scaled | 89.06 | 72.26 | 89.06 | 70.81 | 89.08 | 69.66 | 89.56 | 68.17 | 89.19 | 70.22 |
| MCM | 30.91 | 94.61 | 37.59 | 92.57 | 44.69 | 89.77 | 57.77 | 86.11 | 42.74 | 90.77 |