# OpenReview forum: "Delving into Out-of-Distribution Detection with Vision-Language Representations"
_NeurIPS.cc/2022/Conference — NeurIPS 2022 Accept_

### Official Review · Reviewer_8qRe · 2022-07-11

**Rating:** 6
**Confidence:** 4
**Soundness:** 3 good
**Presentation:** 3 good
**Contribution:** 3 good

**Summary:**

The paper presents an OOD detection approach termed MCM based on vision-language pretrained models, by formatting the OOD detection as a distance-based matching problem. Specifically, they take the advantage of CLIP for distinguishing OOD data from known classes by cross-modal matching, which establishes a powerful cross-modal subspace for metric learning. The author gives some theoretical and empirical evidence to explain the superiority of MCM over cross-entropy-based models.

**Questions:**

1. In Table 2, why do Fort et al. (ViT-B) and MSP (CLIP-B) have the same performance on all metrics? Also, some values should not be bold.
2. What about the variance of the experiments?

**Ethics Review Area:**

["I don’t know"]

**Limitations:**

The paper did not describe limitations.

**Strengths And Weaknesses:**

Strengths:
* The paper is well written and mostly clear.
* The idea of the maximum matching score is simple but effective and shows promising results compared with some baselines.
* The theoretical explanation of softmax scaling seems reasonable and insightful.

Weaknesses:
* Missing comparisons with recent works. The compared works in Table 2 all adopt a Transformer trained on large datasets, a large portion of CNN-based methods is missing, e.g. GradNorm[R1], Energy[R2]. Note that when replacing ResNet with CLIP,  the simple baseline MSP could boost from 63.69/87.59 (reported in [R1]) to 26.65/95.25 on FPR95/AUROC on iNaturalist. I am worried that previous CNN-based methods may also benefit from a strong backbone significantly. The paper lacks a fair comparison with recent CNN-based SOTA methods (MSP and Mahalanobis are somewhat old) using a similar backbone.

* Prompt ensemble. For the same datasets, I guess different prompts may produce unstable performance and thus ensemble works well. I wonder whether the authors have to design new prompts for new datasets.

[R1] On the Importance of Gradients for Detecting Distributional Shifts in the Wild, NIPS 2021.

[R2] Energy-based out-of-distribution detection, NIPS 2020.

---

> ### Author Response · Authors · 2022-08-02
> **Response to Reviewer 8qRe**
>
> >**Q1: Comparisons with recent works**.
>
> Thanks for the suggestion! For the Energy score, please refer to Appendix F.1 for a detailed discussion where we investigate the effectiveness of Energy score based on CLIP. For GradNorm, as suggested, we provide the results as follows. For reference, we also paste the results reported in the original paper (Table 1) [1] based on ResNetv2-101 trained on ImageNet (numbers are FPR95/AUROC).
>
> | Model                   | iNaturalist | SUN         | Places      | Texture     | Average     |
> | ----------------------- | ----------- | ----------- | ----------- | ----------- | ----------- |
> | GradNorm (ResNetv2-101) | 50.03/90.33 | 46.48/89.03 | 60.86/84.82 | 61.42/81.07 | 54.70/86.31 |
> | GradNorm (CLIP-B)       | 68.35/79.53 | 40.74/91.11 | 49.64/87.31 | 48.37/87.51 | 51.77/86.37 |
> | MSP (CLIP-B)            | 40.89/88.63 | 65.81/81.24 | 67.90/80.14 | 64.96/78.16 | 59.89/82.04 |
> | MCM (CLIP-B)            | 32.08/94.41 | 39.21/92.28 | 44.88/89.83 | 58.05/85.96 | 43.55/90.62 |
>
> From the above results, we have several observations: **(1)** given the same feature backbone (CLIP-B), when linear probed on ImageNet-1k, GradNorm indeed improves the average performance compared to the classic MSP score (59.89\% vs. 51.77\% in FPR95); **(2)** GradNorm (CLIP-B) achieves comparable and even better performance compared to GradNorm (ResNetv2-101 trained from scratch on ImageNet) due to better feature representations as a result of large-scale pre-training. For example, the average FPR95 is improved from 54.70\% to 51.77\%; **(3)** Finally, MCM (CLIP-B) still outperform GradNorm by a large margin (43.55\% vs. 54.70\% in FPR95) across most OOD test sets, which is encouraging as MCM is zero-shot and training free.
>
> [1] Huang et al., On the Importance of Gradients for Detecting Distributional Shifts in the Wild, NIPS 2021
>
> >**Q2: Do we have to design new prompts for new datasets**
>
> We agree that prompt design can be an important factor. While we observe prompt ensembling can further improve the performance, it is not a hard requirement.
>
> One interesting finding in our experiments is that, thanks to the powerful pre-trained model, instead of designing different prompts for different datasets,  the default simple prompt "*This is a photo of __*" suffices to achieve promising OOD detection results across different architectures and OOD benchmarks.
>
> Another reason we use the fixed prompt template is for the consideration of fairness. Also, instead of designing prompts for each new dataset, recent advances such as prompt learning [2] might further improve the performance, which we leave as feature work.
>
> [2] Zhou et al., Conditional Prompt Learning for Vision-Language Models, CVPR 2022
>
> >**Q3: In Table 2, why do Fort et al. (ViT-B) and MSP (CLIP-B) have the same performance on all metrics**
>
> Thanks very much for catching the typo! We have updated the table in the manuscript with verified results for Fort et al. (ViT-B). Code will also be released to facilitate reproducibility.
>
> We also post the updated rows here for reference. Numbers are FPR95/AUROC.
>
> | Model               | iNaturalist | SUN         | Places      | Texture     | Average     |
> | ------------------- | ----------- | ----------- | ----------- | ----------- | ----------- |
> | Fort et al. (ViT-B) | 15.07/96.64 | 54.12/86.37 | 57.99/85.24 | 53.32/84.77 | 45.12/88.25 |
> | MSP (CLIP-B)        | 40.89/88.63 | 65.81/81.24 | 67.90/80.14 | 64.96/78.16 | 59.89/82.04 |

---

### Official Review · Reviewer_ZoJQ · 2022-07-12

**Rating:** 6
**Confidence:** 4
**Soundness:** 3 good
**Presentation:** 4 excellent
**Contribution:** 3 good

**Summary:**

The authors proposed Maximum Concept Matching (MCM), a distance-based zero-shot OOD detection method based on aligning visual features with textual concepts, and provided theoretical justifications that softmax scaling used by MCM improves the separability between ID and OOD data for CLIP-based OOD detection. The experiments demonstrate that MCM achieves superior performance on a wide variety of real-world tasks. MCM with vision-language (multi-modal) features outperforms Mahalanobis with visual features only (single-modal) on a hard OOD task. MCM offers several advantages: OOD-agnostic (no information required from OOD data), training-free (no downstream fine-tuning required), and generalizable (one model supports many tasks).

**Questions:**

Can MCM be applicable for non-contrastive vision-language pre-training models?

**Ethics Review Area:**

["I don’t know"]

**Limitations:**

Ablation study on softmax scaling should be given.

**Strengths And Weaknesses:**

Strengths

Originality: the authors adopted softmax scaling, which improves the separability between ID and OOD data.

Quality: MCM achieves superior performance on a wide variety of real-world tasks, compared with the state-of-the-art OOD detection methods. MCM with vision-language (multi-modal) features outperforms Mahalanobis with visual features only (single-modal) on a hard OOD task, with an improvement of 56.60% on FPR95.

Clarity: the manuscript is well-organized and the text is clear and concise.

Significance: The use of the powerful CLIP representation and softmax scaling, enables zero-shot OOD detection. More important, MCM is OOD-agnostic and generalizes to many tasks, making it suitable for real-world applications, where we don’t know OOD labels in prior.


Weaknesses

Originality: the authors proposed MCM inspired by concept match which has been adopted in previous OOD detection work (Mahalanobis, which defines class prototypes based on pure visual embeddings). And multi-modal OOD detection has been practiced in previous OOD detection works yet.

---

> ### Author Response · Authors · 2022-08-02
> **Response to Reviewer ZoJQ**
>
> We sincerely appreciate the reviewer for the positive feedback and insightful comments!
> >**Q1: Comparing MCM with Mahalanobis score**
>
> Conceptually, the Mahalanobis score can be viewed as concept matching based on visual class prototypes. However, there are several key differences:
>
> **(1)** Distributional assumption: Mahalanobis assumes that features follow class-wise multivariate Gaussian distribution, which can be very hard to satisfy for neural network-based features; while MCM based on cross-modal matching does not place any distributional assumption and thus is more generalizable.
>
> **(2)** Sample efficiency: estimating the covariance matrix for the Mahalanobis score can be challenging for high-dimensional feature space. For example, for error $\epsilon \in (0,1)$, under mild conditions, we require a sample size of $O(\epsilon^{-2}d\log d)$ to obtain a good estimate (Remark 5.6.2 in [1]), where $d$ is the dimension of features. This can be challenging in practice. In contrast, MCM directly alleviates the need for estimating the covariance matrix.
>
> **(3)** Visual prototypes vs. textual prototypes: compared to textual labels such as “cat”, cat images usually contain background and potentially spurious attributes (Appendix C), and as a consequence, when the average of visual features are used as prototypes for the class “cat” (as in Mahalanobis score), it may inherit spurious information. The performance of Mahalanobis score and MCM for spurious OOD samples are reported in Section 4.2 (Table 3).
>
> [1] Vershynin, Roman. High-dimensional probability: An introduction with applications in data science. Cambridge university press, 2018.
>
> >**Q2: Comparing MCM with prior works on multi-modal OOD detection**
>
> To the best of our knowledge, two recent works [2,3] also utilize multi-modal information for OOD detection.
>
> In **L253 - 275** (page 7), we have compared MCM with two recent works utilizing multi-modal information for OOD detection, with extended implementation details in **Section E (appendix)**. There exist fundamental limitations for both works. For example, Fort et al. [2] assume prior knowledge for test OOD samples, while ZO-CLIP[3] cannot guarantee the generated labels are non-overlapping with ID labels. In contrast, MCM is simpler, more efficient (no need to train a decoder), and shows promising results compared to ZO-CLIP[3] across a wide range of benchmarks. On the other hand, compared to Fort el al., MCM does not rely on any prior information and therefore can be generally applied.
>
> > **Q3: Ablation study on softmax scaling**
>
> We really appreciate the suggestion! Please refer to our response to Q3 of Reviewer rjjS. The ablation study is also added in Appendix H.
>
> > **Q4: Can MCM be applicable for non-contrastive vision-language pre-training models**
>
> Nice question! Our MCM score is built on top of the aligned cross-modal features, and it is designed to scale aligned cross-modal features to magnify the difference between ID and OOD. Therefore, the MCM score should also be applied to models pre-trained with multi-modal alignment prediction objectives, including recent works such as RegionCLIP [4], GroupViT [5], etc. We will extend the experiments (with more recent pre-trained models) in the revised version.
>
>
>
> [2] Fort et al., Exploring the Limits of Out-of-Distribution Detection, NIPS 2021
>
> [3] Esmaeilpour et al., Zero-Shot Out-of-Distribution Detection Based on the Pre-trained Model CLIP, AAAI 2022
>
> [4] Zhong et al., RegionCLIP: Region-based Language-Image Pretraining, CVPR 2022.
>
> [5] Xu et al., GroupViT: Semantic Segmentation Emerges from Text Supervision, CVPR 2022

---

### Official Review · Reviewer_rjjS · 2022-07-12

**Rating:** 6
**Confidence:** 3
**Soundness:** 3 good
**Presentation:** 3 good
**Contribution:** 2 fair

**Summary:**

This paper proposed a new out-of-distribution (OOD) detection method based on pretrained vision-language model. Unlike previous works, this work relies on pretrained models like CLIP to infer whether a given image is in or out of the target distribution by matching the visual features to a set of concept embeddings. To mitigate the score overlap between in-domain and out-of-domain data, the authors proposed a simple but effective softmax-scaling to recalibrate the matching scores between images and texts. According to the experimental results, the proposed method called Maximum Concept Matching (MCM) achieves superior performance to previous works on a variety of benchmarks and settings. It does not need any finetuning but can be directly applied to different datasets and label spaces in a zero-shot manner thanks to the zero-shot capacity learned by vision-langauge models.

**Questions:**

Questions:

1. What is the performance after finetuning CLIP models for OOD detection? It would be better to show the performance after finetuning on downstream datasets.

2. It is still not clear to me why vision-language pretrained model with simple softmax scaling can outperform previous work significantly. There is not much explanation in the paper about this.

3. The authors should report the performance without softmax scaling for all datasets and evaluations. The role of softmax scaling is not clear to me.

Suggestions:

1. I do not think the claimed contributions in lines 155-168 are from the proposed MCM method. It mainly inherits from the off-the-shelf CLIP models. All methods that use vision and language encoders in CLIP are supposed to possess such properties.

2. The authors should have more discussions about the contribution of proposing softmax scaling, which I think is an incremental and normal technique.



**Limitations:**

The authors did not discuss the limitations and societal impact in their submission.

**Strengths And Weaknesses:**

[Strenghts]

1. This paper introduced a simple yet effective OOD detection method called MCM based on multi-modal pre-trained models. This is one of the first works that leverage vision-language multi-modal models for OOD, in contrast to previous single-modal methods.

2. Based on the vision-language pretrained model like CLIP, the authors further proposed a simple softmax scaling to re-calibrate the matching scores to magnify the difference between in-distribution and out-of-distribution samples.

3. Comprehensive experiments demonstrate that the proposed MCM outperforms previous work proposed by Fort et al. and MSP. The superiority is universal on both small-scale datasets and large-scale ones line ImageNet-1K. The authors further demonstrated that MCM can benefit hard OOD detection very well.

[Weaknesses]

1. The technical contribution in this paper is limited. First of all, using softmax on the classification logits is natural and straightforward as shown in MSP and other works. Despite the authors gave proof that softmax scaling can improve the OOD detection when using clip-based models, I think this change is incremental and should be considered as a baseline method.

2. The proposed MCM is one of the first works that leveraged vision-language pretrained models. However, I do think the claims made in lines 155-168 are exaggerated. Most of the claimed merits for MCM are due to vision-language pretrained CLIP, such as generalization ability, training-free and scalable.

3. In Fig.4, the authors showed the effect of softmax scaling and the corresponding FPR at different settings. For completeness, the authors should report the results on all used datasets for the proposed method without softmax scaling.

4. The authors merely reported the performance on zero-shot settings, it would be better to also report the performance after finetuning. It is not clear whether after finetuning the OOD detection performance will be further improved or not.

5. It is not clear in the paper why the vision-language model achieves much better zero-shot OOD detection than the single-modal method. The authors comapred with Maha in lines 289-307. However, why text features as the prototypes are supposed to be better than visual features as the prototypes?

---

> ### Author Response · Authors · 2022-08-02
> **Response to Reviewer rjjS**
>
> > **Q1: The role of softmax scaling**
>
> Great question! We provided an expansive explanation in **L113-L152, Page 4**. Despite its simple form, our insights behind using softmax scaling for the CLIP-like model is new, and *contrary* to the findings based on the cross-entropy loss. In fact, since MSP was introduced as a baseline, it took the community a few years of research efforts to realize that logit-based scores without softmax scaling are more effective for models trained with cross-entropy loss. In light of this, we are very deliberate and careful in bringing softmax scaling to the picture, for CLIP-like models.
>
> In particular, applying softmax to cross-modal cosine similarity is *fundamentally different from the softmax score derived from a model trained with cross-entropy loss*, which inherently maximizes the posterior $p(y|\mathbf{x})$ for the ground-truth label, and minimizes the probability for other labels. Unlike CLIP-like models, logit scores displaying uniformity would be heavily penalized by the CE loss.
>
> As a result, the logit score corresponding to the ground truth label can already be significantly higher than other labels. Applying softmax on the logit scores can exacerbate overconfident predictions, and reduce the separability between ID and OOD data. On the contrary, for CLIP-like models, applying softmax helps sharpen the uniform-like inner product scores, and increases the separability between ID and OOD data. We provide theoretical justifications in **Thm 3.1** that softmax scaling indeed improves the separability between ID and OOD data for CLIP-based OOD detection, with full proof in the appendix.
>
> With that being said, we agree with the reviewer that the simplicity and effectiveness of our method make it a proper baseline for future works on multi-modal OOD detection. Moreover, we hope our theoretical insights are valuable to the community.
>
> > **Q2: Merits of MCM**
>
> For OOD detection, most prior works focus on a single modality & task-specific models, leaving rich opportunities for multi-modal features untapped. Our paper aims to provide a timely investigation and highlight the compelling advantages of OOD detection with aligned multi-modal features. We hope MCM can serve as a springboard and a simple baseline for future works on OOD detection in this direction. *The highlighted merits are mostly positioned w.r.t. existing OOD detection literature*, though we agree that the merits are indispensable from the CLIP model. We will clarify this in the revised version.
>
> > **Q3: OOD detection performance without softmax scaling**
>
> Thanks very much for the suggestions! We have added an ablation study in Appendix H. The results *without* softmax scaling (i.e. using the maximum cosine similarity) are shown below based on CLIP-B/16 (numbers are FPR95/AUROC).
>
> | ID/OOD        | iNaturalist | SUN         | Places      | Texture     | Average     |
> | ------------- | ----------- | ----------- | ----------- | ----------- | ----------- |
> | ImageNet-10   | 41.15/95.74 | 33.31/96.45 | 30.6/96.6   | 34.57/96.44 | 34.91/96.31 |
> | ImageNet-20   | 14.88/97.91 | 14.74/97.86 | 13.49/97.84 | 13.42/97.39 | 14.13/97.75 |
> | ImageNet-1k   | 56.59/90.27 | 64.28/87.60 | 57.23/87.76 | 84.2/73.5   | 65.57/84.78 |
> | Stanford-Cars | 0.00/100.00 | 0.03/99.99  | 0.29/99.94  | 0.00/100.0  | 0.08/99.98  |
> | Food-101      | 0.65/99.83  | 0.21/99.92  | 0.52/99.86  | 9.31/97.15  | 2.67/99.19  |
> | Oxford-Pet    | 0.00/99.99  | 0.04/99.98  | 0.19/99.95  | 0.27/99.92  | 0.12/99.96  |
>
> For easy tasks such as Food-101, Stanford-Cars, and Oxford-Pet as ID, the performance of maximum cosine similarity score is similar to MCM (Tables 1 and 2). However, for more challenging tasks such as ImageNet-20 and ImageNet-1k, MCM significantly outperforms that without softmax scaling. For example, the average FPR95 is improved by 11.33\% on ImageNet-20 and 22.02\% on ImageNet-1k, which highlights the necessity of a proper scaling function for CLIP-based OOD detection.

---

> > ### Author Response · Authors · 2022-08-02
> > **Author Response (Continued)**
> >
> > > **Q4: Performance with fine-tuning CLIP models**
> >
> > In Table 2, we compared MCM with a range of recent methods that require fine-tuning such as Fort et al. (based on ViT), MOS (based on BiT), and MSP (fine-tuned the ViT model in CLIP, same backbone as ours). Compared to the baselines, we show that MCM remains very competitive without fine-tuning on ImageNet-1K.
> >
> > During our exploration, we did consider fine-tuning the entire backbone. However, we find that **(1)** simply fine-tuning both text and image encoders with the CLIP loss does not lead to consistent improvement for OOD detection as fine-tuning the large feature backbone without special optimization strategies can distort aligned cross-modal features learned during pre-training; **(2)** only fine-tuning the image encoder also does not yield consistent improvements compared to linear-probing.
> >
> > Our findings also echo a conclusion in a recent paper [1] on OOD generalization that shows fine-tuning the feature backbone leads to worse accuracy than linear-probing when the pre-trained features are good, and the distribution shift is large.
> >
> > >**Q5: Comparison with Mahalanobis score. Why text features as the prototypes are supposed to be better than visual features as the prototypes?**
> >
> > Thanks for the comment! Please see the first answer below in response to Reviewer ZoJQ.
> >
> > [1] Kumar et al., Fine-Tuning can Distort Pretrained Features and Underperform Out-of-Distribution, ICLR 2022

---

### Official Review · Reviewer_Ekfu · 2022-07-13

**Rating:** 6
**Confidence:** 4
**Soundness:** 3 good
**Presentation:** 3 good
**Contribution:** 3 good

**Summary:**

This paper presents an out-of-distribution(OOD) detection method for the image classification models based on the CLIP model. The authors propose Maximum Concept Matching which detects OOD by aligning visual features with textual concepts. Experiments show that this simple yet effective method can outperform many methods. Since the method does not need any training, it can be easily applied to different datasets with minimum effort.

**Questions:**



**Limitations:**

No limitations are discussed by the authors.

**Strengths And Weaknesses:**

Strengths:
- The method is simple and can work well without bells and whistles.
    - There're four compelling advantages: generalizable to many tasks; OOD-agnostic; training-free; scalable.
- The authors justify their design with theoretical proof.

Weaknesses:
- Although the definition of MCM in eq. 1 is intuitive, there could be many ways to detect uniformity of the cosine similarities. For example, entropy after softmax scaling, the variance of the cosine similarities, exp(max - second_max). More options should be compared in the experiments.
- It will be great if the authors can try more different CLIP models like resnet CLIP, and some community open sourced CLIP-like models. Would it be possible that it is actually the vision transformer or the training data of CLIP to be the root cause of strong OOD detection performance?
- Wrong fact: the authors claim that CLIP uses a temperature of 0.07. However, the temperature of CLIP is initialized as 0.07 and learned during training.

Final rating:
The authors' rebuttal addresses my questions very well. I will keep my original rating.

---

> ### Author Response · Authors · 2022-08-02
> **Response to Reviewer Ekfu**
>
> > **Q1: Alternative scoring functions**
>
> We thank the reviewer for the positive feedback and insightful comments! In this work, we use the softmax function with temperature to improve the separability between ID and OOD due to its simplicity and theoretical guarantees (**Thm 3.1**) for CLIP-like models. As suggested, we report the results for the large-scale benchmark ImageNet-1K blow (numbers are FPR95/AUROC):
>
> | Score                                   | iNaturalist | SUN         | Places      | Texture     | Average     |
> | --------------------------------------- | ----------- | ----------- | ----------- | ----------- | ----------- |
> | MCM                                     | 32.08/94.41 | 39.21/92.28 | 44.88/89.83 | 58.05/85.96 | 43.55/90.62 |
> | Entropy after softmax (best $\tau$)     | 44.63/83.76 | 50.52/81.53 | 52.75/79.00 | 64.15/75.94 | 53.01/80.06 |
> | Variance of cosine similarity           | 85.45/65.41 | 70.00/80.32 | 75.29/76.14 | 79.79/71.60 | 77.63/73.37 |
> | Exp(max - second max cosine similarity) | 88.90/71.48 | 90.25/67.86 | 89.39/69.61 | 90.25/67.86 | 89.47/69.87 |
>
> For the Entropy score, we select $\tau \in$ \{0.005, 0.01, 0.1, 1, 10\} and report the performance with the best $\tau$.  We observe that on the large-scale benchmark ImageNet-1k (ID), MCM still gives **the most promising** results compared to the other three alternative scores. We have included the ablation study comparing MCM with all the scores suggested in Appendix F.2.
>
> > **Q2: ResNet-CLIP vs. ViT-CLIP**
>
> Great suggestion! We would like to discuss the cause of strong OOD detection performance for CLIP-like models with the MCM score from the following two aspects:
>
> **(1)** It is true that CLIP-like pre-training models boost the performance of OOD detection. Large-scale pre-training is beneficial for transferrable representation learning [1]. Therefore, the task of OOD detection benefits from a powerful pre-trained model. As suggested, to illustrate the effectiveness of MCM score on ResNet-based CLIP, we use RN50x4 (178.3M) which shares  similar number of parameters as CLIP-B/16 (149.6M). The OOD detection performance of MCM for the large-scale benchmark ImageNet-1k (ID) is shown below (numbers are FPR95/AUROC):
>
> | Model     | iNaturalist | SUN         | Places      | Texture     | Average     |
> | --------- | ----------- | ----------- | ----------- | ----------- | ----------- |
> | CLIP-B/16 | 32.08/94.41 | 39.21/92.28 | 44.88/89.83 | 58.05/85.96 | 43.55/90.62 |
> | RN50x4    | 44.51/91.51 | 35.11/92.84 | 43.74/89.60 | 57.73/85.93 | 45.27/89.97 |
>
> It can be seen that MCM still shows promising results with ResNet-based CLIP models, and the performance is comparable between RN50x4 and CLIP-B/16  (89.97 vs. 90.62 in AUROC). We have added this ablation to Appendix G.
>
> **(2)**  We would like to highlight the importance of a proper detection score. Given the most powerful CLIP model (CLIP-L/14), directly using cosine similarity *without softmax scaling* for OOD detection only yields an average FPR95 of 60.92% for ImageNet-1k (ID), significantly worse than MCM (37.19%). In addition, a recent work (ZO-CLIP) [2] also uses CLIP for OOD detection. Given the same encoder, we enhance ZO-CLIP with a stronger decoder and a filter module (Appendix E). MCM still consistently outperforms ZO-CLIP (Figure 3) across the 4 benchmark OOD test sets, which also suggests that a powerful feature representation alone is insufficient for strong OOD detection performance.
>
> > **Q3: The temperature of CLIP during pre-training.**
>
> Thanks very much for pointing this out. We have revised the description in the updated manuscript.
>
> [1] Radford et al., Learning Transferable Visual Models From Natural Language Supervision, ICML 2021
>
> [2] Esmaeilpour et al., Zero-Shot Out-of-Distribution Detection Based on the Pre-trained Model CLIP, AAAI 2022

---

### Author Response · Authors · 2022-08-02
**Summary of response -- thanks to all reviewers for thorough and insightful feedback**

We thank all the reviewers for their time, insightful suggestions, and valuable comments.  We are glad that **ALL** reviewers appreciate our work and find our method simple and effective, one of the first works that leverage vision-language multi-modal models for OOD (rjjS), with comprehensive experiments on small-scale, large-scale, and hard OOD detection tasks where MCM demonstrate promising results (rjjS, ZoJQ, 8qRe).  We are also encouraged that reviewers find our theoretical explanations of softmax scaling reasonable (Ekfu) and insightful (8qRe); the paper is well-organized, clear, and concise (ZoJQ, 8qRe).

We respond to each reviewer's comments in detail below. We have also revised the manuscript according to reviewers' suggestions, and we believe this makes our paper much stronger.  The main changes we made include:

- In Appendix F.2, we add an ablation study comparing MCM with other scaling functions on cosine similarities (as Reviewer Ekfu suggested)
- In Appendix G, we add an ablation study on the performance of MCM with ResNet-based CLIP models.
- In Appendix H, we add an ablation study on the performance without softmax scaling for all tasks.

*In the revised manuscript, we have marked the revisions in blue.

---

> ### Public Comment · ~Jing_Li31 · 2023-10-07
> **How to decide the value of lambda in the OOD detection function?**
>
> The paper goes "lambda is chosen so that a high fraction of ID data (e.g., 95%) is above the threshold". MCM is claimed as "training-free". However,  A labeled training set is still necessary if lambda is chosen in the way above. Am I right?

---

> > ### Public Comment · Authors · 2023-10-07
> > **How to decide the value of lambda**
> >
> > Hi! Thanks for the question. Selecting $\lambda$ to achieve a true positive rate of 95% is a standard practice for OOD detection methods when evaluating threshold-dependent metrics like FPR95 (FPR@TPR95). This choice is about evaluation criteria, not an intrinsic aspect of the detection method. In contrast, threshold-free metrics like AUROC don't require a specific $\lambda$.

---

> > > ### Public Comment · ~Jing_Li31 · 2023-10-08
> > > **still confused**
> > >
> > > Hi, thanks for answering and explaining the evaluation criteria. When you predict a test sample as ID or OOD, you have to compare its maximum concept matching score with the value of lambda, right? So, I think an exact lambda value is necessary. My question is how to obtain this lambda value?

---

> > > > ### Public Comment · Authors · 2023-10-10
> > > > **How to obtain lambda in practice**
> > > >
> > > > Yes, a typical OOD detector consists of an OOD score and a threshold $\lambda$.  A common way to determine $\lambda$ (note that there are alternative ways in the literature) is to use ID data so that the true positive rate is at 95%, which corresponds to the metric FPR95.  See L108-L117 https://github.com/deeplearning-wisc/MCM/blob/main/utils/detection_util.py.

---

> > > > > ### Public Comment · ~Jing_Li31 · 2023-10-10
> > > > > **Thank you for being patient in explaining.**
> > > > >
> > > > > Thanks~

---

### Meta-Review · Area_Chair_w6nm · 2022-08-26

**Recommendation:** Accept
**Confidence:** Certain

**Metareview:**

This paper presents an interesting and novel try at using vision-language multi-modal models for OOD tasks. The experiments are sufficiently validated on diverse OOD datasets. Besides, the theoretical explanations of softmax scaling are quite insightful. All reviewers give positive scores. During the discussion phase, the authors have accordingly added more ablation studies and comparisons. Despite some reviewers raising concerns about the possible limited novelty of using softmax scaling for the CLIP-like model. The authors give sufficient discussions and provide theoretical justifications, which will inspire the community. The meta-reviewers thus recommend accepting it.

**Award:**

No

---

### Decision · Program_Chairs · 2022-09-14

Accept